# Spidroin N-terminal domain forms amyloid-like fibril based hydrogels and provides a protein immobilization platform

Tina Arndt [1], Kristaps Jaudzems [2], Olga Shilkova [1], Juanita Francis[1], Mathias Johansson [3], Peter R. Laity[4], Cagla Sahin [5], Urmimala Chatterjee [1], Nina Kronqvist [1], Edgar Barajas-Ledesma[5], Rakesh Kumar[1], Gefei Chen [1], Roger Strömberg[1], Axel Abelein [1], Maud Langton [3], Michael Landreh [5], Andreas Barth[6], Chris Holland [4], Jan Johansson[1] & Anna Rising [1,7] ✉

Recombinant spider silk proteins (spidroins) have multiple potential applications in development of novel biomaterials, but their multimodal and aggregation-prone nature have complicated production and straightforward applications. Here, we report that recombinant miniature spidroins, and importantly also the N-terminal domain (NT) on its own, rapidly form self-supporting and transparent hydrogels at 37 °C. The gelation is caused by NT α-helix to β-sheet conversion and formation of amyloid-like fibrils, and fusion proteins composed of NT and green fluorescent protein or purine nucleoside phosphorylase form hydrogels with intact functions of the fusion moieties. Our findings demonstrate that recombinant NT and fusion proteins give high expression yields and bestow attractive properties to hydrogels, e.g., transparency, cross-linker free gelation and straightforward immobilization of active proteins at high density.

Spiders have up to seven different sets of silk glands each producing a specific type of silk. All seven silks are composed of spider silk proteins (spidroins) that are up to ~6000 residues long and contain an extensive central repetitive region capped by globular N- and C-terminal domains (NT and CT)[1,2]. The most extensively studied silk type, major ampullate, is produced by the major ampullate gland. In this gland, a single layered epithelium synthesizes the spidroins and secretes them into the gland lumen where they are stored in a soluble form (dope) at extremely high concentrations (30–50% w/v)[3,4]. The organization and conformation of the major ampullate spidroins in the gland have been debated, but most experimental evidence points towards an overall helical and/or random coil conformation and the existence of micelles or flake-like structures[5–10]. While the repetitive region mediates the mechanical properties of the silk fiber by forming β-sheet nanocrystals and amorphous structures[11–15], the terminal domains control silk formation by responding to altered conditions along the silk gland[16–19]. The terminal domains are evolutionary conserved and their function is likely universal in all spidroins[2,20,21]. During passage through the gland, the spidroins experience a drop in pH from around 7.6 to < 5.7[16] and increased shear and extensional forces mediated by moving through the progressively narrowing duct[22]. In solution, CT is an α-helical constitutive parallel dimer[17] but in response to low pH and shear forces CT undergoes unfolding and β-sheet conversion[16,17], possibly triggering β-sheet transition of the repetitive

[1]Department of Biosciences and Nutrition, Karolinska Institutet, Neo, Blickagången 16, Huddinge 141 52, Sweden. [2]Department of Physical Organic Chemistry, Latvian Institute of Organic Synthesis, Riga LV-1006, Latvia. [3]Department of Molecular Sciences, Swedish University of Agricultural Sciences, Uppsala 750 07, Sweden, Box 7015. [4]Department of Materials Science and Engineering, The University of Sheffield, Sir Robert Hadfield Building, Mappin Street, Sheffield S1 3JD, UK. [5]Department of Microbiology, Tumor and Cell Biology, Karolinska Institutet, Solnavägen 9, 171 65 Solna, Sweden. [6]Department of Biochemistry and Biophysics, The Arrhenius Laboratories for Natural Sciences, Stockholm University, 10691 Stockholm, Sweden. [7]Department of Anatomy, Physiology and Biochemistry, Swedish University of Agricultural Sciences, Uppsala 750 07, Sweden. ✉e-mail: anna.rising@ki.se

region[16]. NT is monomeric under conditions that reflect those in the gland lumen and mediates solubility to the spidroins, but at decreased pH, protonation of a series of carboxylate side chains leads to NT dimerization with a pKa of around 6.5, which stabilizes NT and locks the spidroins in large networks[16,18]. Thus, NT plays a key role in the silk formation process by transitioning from a monomer in the dope to a dimer in the fiber[23–25]. NT remains highly soluble and helical at all conditions investigated to date[16,18–20,26–29], which has inspired its development into a solubility enhancing tag for heterologous protein production[30].

A recombinant mini-spidroin that is composed of an NT, a short repeat region, a CT and a His$_6$-tag for purification (His-NT2RepCT), is as soluble as native spider silk proteins in aqueous buffers and recapitulates important features of native spider silk dope[25,31]. The His-NT2RepCT can be spun into continuous fibers using biomimetic set-ups where the soluble dope at pH 8 is extruded into an aqueous pH 5 bath[25,32–35]. Bioreactor fermentation of *Escherichia coli* expressing His-NT2RepCT and subsequent down-stream processing result in a yield of >14 g/l after purification[34]. Both the high yields of His-NT2RepCT, its high solubility and proper response to acidic conditions have been attributed to the NT[23,25,34].

Herein, we report on rapid formation of transparent hydrogels from recombinant spidroins, including NT alone, by incubation of the protein solution at 37 °C. By using Thioflavin T (ThT) fluorescence,

Fourier Transform Infrared (FTIR) spectroscopy, nuclear magnetic resonance (NMR) spectroscopy and transmission electron microscopy (TEM), we find that NT and miniature spidroins undergo a structural transition into β-sheets and amyloid-like fibrils upon gel formation. Furthermore, fusion proteins of NT and green fluorescent protein (GFP) or purine nucleoside phosphorylase (PNP) form hydrogels with intact functions of the fusion moieties. The combination of high-yield expression in heterologous hosts and rapid formation of hydrogels under physiological conditions open the possibility of economically feasible production of hydrogels with designed functions.

## Results

### Temperature-induced hydrogel formation of recombinant spidroins

In contrast to most reported recombinant spidroins[36], His-NT2RepCT is stable in Tris-HCl buffers at pH 8 and can be concentrated to 500 mg/ml without precipitating[25]. Hence, we were surprised to find that this protein rapidly forms optically transparent, self-supporting hydrogels when incubated at 37 °C (Fig. 1b–d). Further investigation showed that gelation of His-NT2RepCT occurs at a wide range of protein concentrations (10–300 mg/ml) and the concentration correlates inversely with time for gelation (Fig. 1c and Supplementary Fig. 1). To elucidate which parts of His-NT2RepCT mediate hydrogel formation, we next studied each domain in isolation and in different

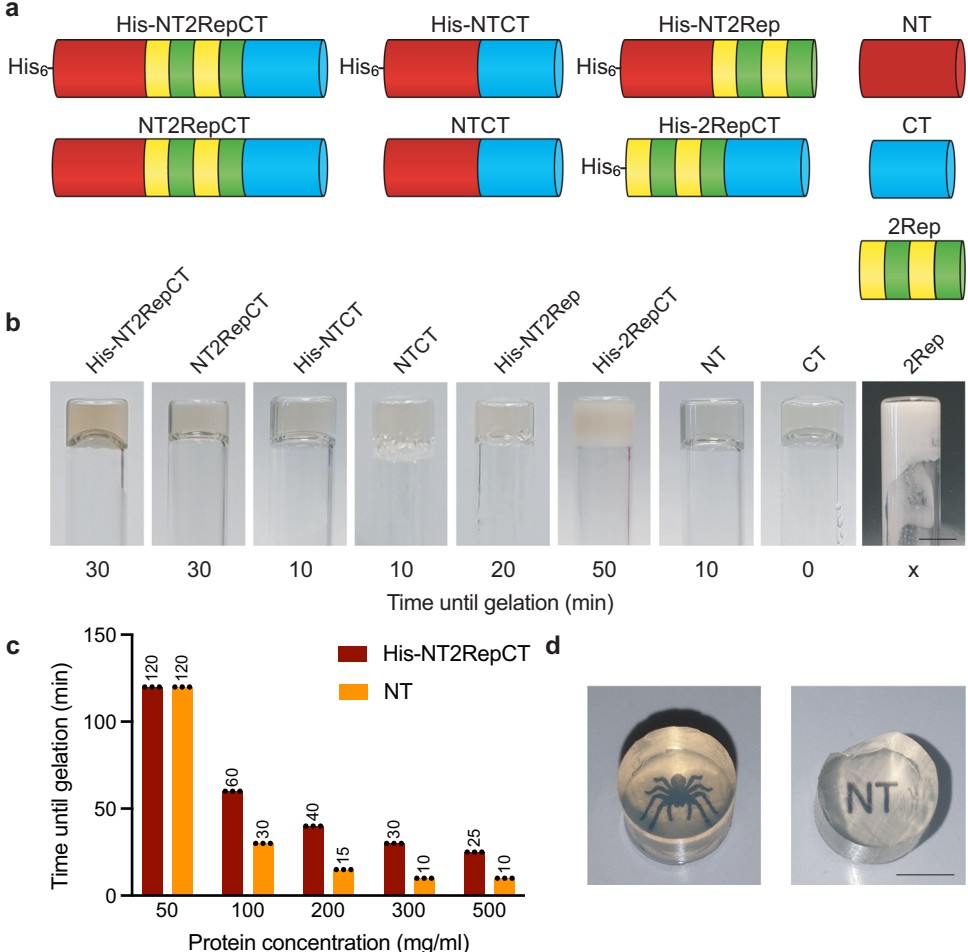

**Fig. 1 | Temperature-induced hydrogel formation of His-NT2RepCT and its different domains. a** Schematic presentation of the different spidroin constructs investigated herein. **b** Gelation times of the different recombinant spidroins (300 mg/ml) at 37 °C as tested by vial inversion. CT gelled immediately without incubation (< 300 mg/ml) and 2Rep precipitated (300 mg/ml, scale bar is 5 mm). **c** Time until gelation of His-NT2RepCT and NT at the indicated protein concentrations at 37 °C. **d** Photographs of His-NT2RepCT and NT hydrogels with a printed spider and letters "NT" underneath, respectively (both 200 mg/ml, scale bar is 5 mm).

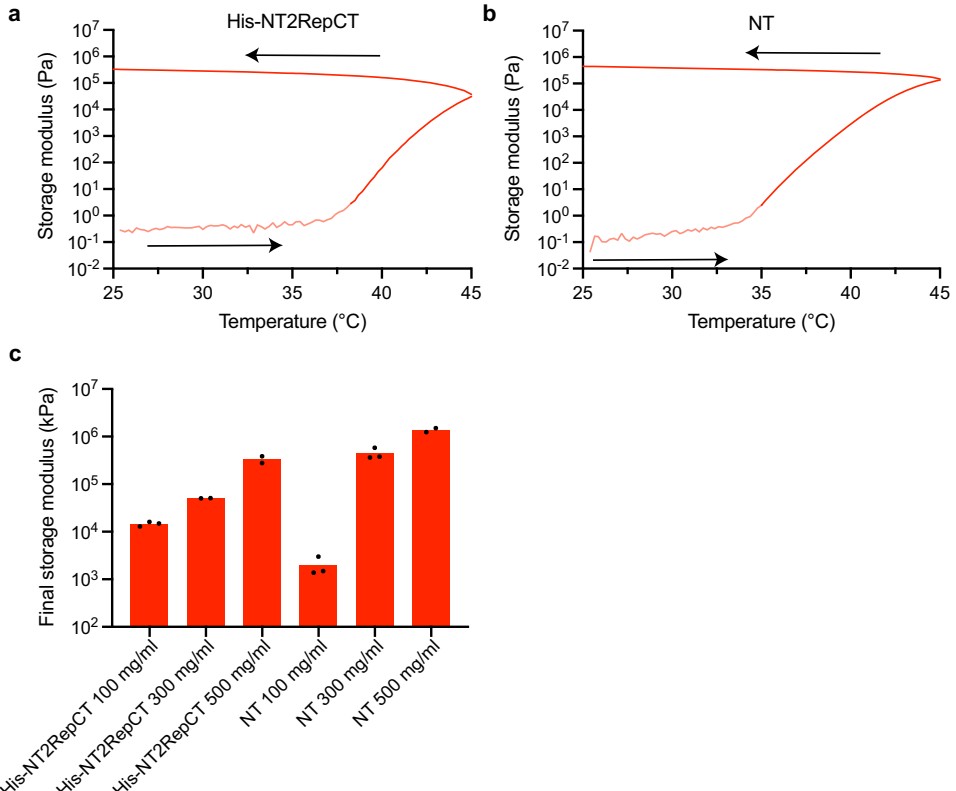

**Fig. 2 | Rheological analyses of His-NT2RepCT and NT gelation. a** Temperature ramp during oscillatory measurement of His-NT2RepCT (300 mg/ml) and **b** NT (300 mg/ml). Arrows indicate temperature course, lighter shaded portions of the storage modulus data depict testing below the manufacturers stated lower torque values for the instrument, accounting for the increased noise. **c** Final storage modulus of His-NT2RepCT and NT after temperature ramps (100, 300, and 500 mg/ml). All modulus readings were taken at 0.1 Hz.

combinations by vial inversion test (Fig. 1a, b). All recombinant spidroin parts tested formed gels in less than 1 h (at protein concentrations of 300 mg/ml) except 2Rep which precipitated (Fig. 1b). This shows that both NT and CT in isolation, in combination or linked to a repetitive part, can form gels at 37 °C, and that the $His_6$-tag does not affect the process to any significant extent. The finding that NT on its own could form gels was unexpected considering the common notion that NT is a highly soluble and stable protein and that previous reports of recombinant spidroin hydrogels have attributed the gelation effect to conformational changes in the repetitive region and/or CT (Supplementary Table 1)[37–39]. Notably, NT formed gels already within 10 min at ≥ 300 mg/ml (Fig. 1c). Vial inversion tests of NT at different concentrations showed that at > 50 mg/ml the NT solution gels faster than His-NT2RepCT at the corresponding concentration (w/v, Fig. 1c).

The hydrogels formed from the different recombinant spidroins had slightly different colors and showed different degree of transparency as judged by the naked eye (Fig. 1b). The NT gels were exceptionally clear while others became opaque. His-NT2RepCT and NT gels that were cast in cylindrical tubes could be removed intact from the molds (Fig. 1d).

To test if native spider silk dope also gels under the conditions now found to induce gelation of recombinant spidroins, dope was collected from the major ampullate glands of a Swedish bridge spider (*Larinioides sclopetarius*). The dope was kept in 20 mM Tris-HCl buffer at a concentration of 50 mg/ml (based on measured dry weight), but gelation was not observed during an incubation period of 21 days at 37 °C (Supplementary Fig. 2a).

Moving towards a quantitative appreciation of these gels, rheological measurements can be used to study both the gelation process and determine overall mechanical properties. Specifically, monitoring the storage (elastic) modulus during temperature ramps can give information on the gelling temperature as well as the viscoelastic properties of the dopes. Temperature ramp experiments (using a rate of 1 °C/min, 25–45 °C in accordance with previous investigations using native silkworm silk dope)[40,41] showed that the storage moduli of both His-NT2RepCT and NT solutions increased when the temperature increased (Fig. 2 and Supplementary Fig. 3). Notably, the modulus of NT began to rise at lower temperatures compared to His-NT2RepCT which agrees with the faster gelation times seen for NT vs. His-NT2RepCT when incubated directly at 37 °C (Fig. 1). Upon a subsequent decrease in temperature the storage modulus did not return to lower values, and remained above the loss modulus (see Supplementary Fig. 3) indicating a thermally irreversible, stable gel formation. After gelation, the final storage modulus was between 15 and 330 kPa for 100–500 mg/ml His-NT2RepCT hydrogels and 2–1400 kPa for NT hydrogels (100–500 mg/ml), (Fig. 2 and for full ramp data see Supplementary Fig. 3).

## Hydrogel formation is associated with formation of ß-sheets and amyloid-like fibrils mainly of NT

As a potential method to investigate conformational changes associated with gelation, we recorded FTIR spectra of His-NT2RepCT and NT before and after gelation at 37 °C (Fig. 3a, b). As expected, spectra of His-NT2RepCT and NT solution were consistent with proteins presenting α-helical/random coil secondary structures with a prominent band at $1645 \text{ cm}^{-1}$. Gelation generated two shoulders in the amid I band at around $1617 \text{ cm}^{-1}$ and $1695 \text{ cm}^{-1}$ for both hydrogels (Fig. 3a, b), suggesting the formation of antiparallel β-sheet structures. These changes can also clearly be seen in the respective second derivative spectra and in difference spectra of gel formation (Supplementary Fig. 4b). Both β-sheet bands were more pronounced for NT than for

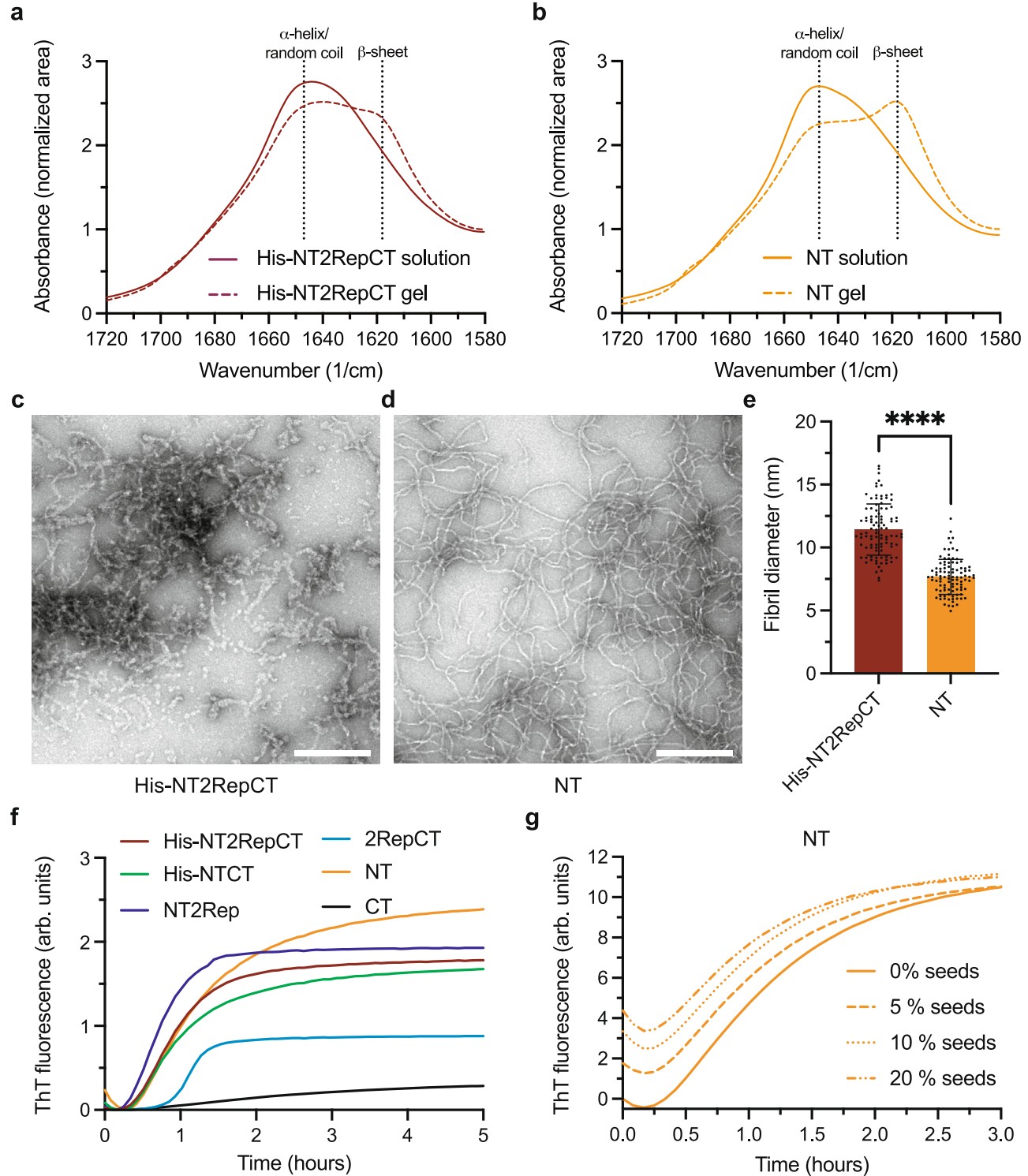

**Fig. 3 | Structural changes during hydrogel formation. a** FTIR absorption spectra of His-NT2RepCT and **b** NT (both at 500 mg/ml) before (solution) and after (gel) incubation at 37 °C. **c** TEM image of a resuspended 50 mg/ml NT2RepCT and **d** NT gel. Scale bars are 200 nm. **e** Fibril diameters of His-NT2RepCT and NT hydrogels. *n* = 100 fibrils measured, *p* < 0.0001. Error bars show standard deviation. Center of the error bar is the mean. Unpaired *t*-test (two-tailed) was used for statistical analysis. **f** ThT fluorescence of different recombinant spidroins (100 mg/ml) at 37 °C without shaking. **g** Seeding experiments of NT (100 mg/ml) with 0, 5, 10 and 20% seeds from a 100 mg/ml NT gel.

His-NT2RepCT, suggesting a higher overall β-sheet content in NT hydrogels than in NT2RepCT hydrogels.

Analysis of the gels by transmission electron microscopy (TEM) revealed that the hydrogels are composed of amyloid-like fibrils (Fig. 3c, d). NT formed fibrils that were elongated, thin (5–12 nm in diameter) and unbranched whereas His-NT2RepCT fibrils were shorter in length and had a significantly wider diameter (7–16 nm), (Fig. 3e). These findings led us to follow the fibrillation kinetics by using a Thioflavin T (ThT) assay. For all recombinant spidroins, the fluorescence signal increased when the samples were incubated at 37 °C (Fig. 3f, Supplementary Fig. 5a). In line with this finding, microscopy of NT and His-NT2RepCT under gel-forming conditions showed a

homogeneous increase in ThT fluorescence, with no discernable local accumulation of ThT-positive aggregates (Supplementary Fig. 5b, c). Formation of ThT positive fibrils occurred without concomitant increase in turbidity for NT and His-NTCT (Supplementary Fig. 5d) which means that the fibrillar network in the gels can form without compromising the transparency of the gels. Seeding by addition of small amounts of preformed fibrils can greatly accelerate fibril formation of some amyloids[42–44], but addition of 5, 10 or 20 % (w/w) NT hydrogels to NT solutions did not give any pronounced seeding effect (Fig. 3g). Possibly, this could be due to that the fibrils in the hydrogels are relatively fixed and may be inaccessible for acting as seeds.

The unexpected behavior of the recombinant spidroins at elevated temperature prompted further investigation with nuclear magnetic resonance (NMR) spectroscopy to dissect the conformational changes associated with gel formation. Solution NMR spectra of His-NT2RepCT recorded over time at 37 °C, revealed that CT remains partly folded while NT and 2Rep signals disappear (Fig. 4a), which suggests that mainly the NT and 2Rep parts drive hydrogel formation of His-NT2RepCT. The CT signals are also attenuated to ~20% of their original intensity, which indicates that CT too is largely immobilized and incorporated into the hydrogel structure. For the smaller fraction of CTs that remain as mobile as in the pre-incubation sample and therefore visible by solution NMR, the spectra lack signals of the first ~10 structured residues, probably due to hindered motion from immobilization of the attached His-NT2Rep part. Solid-state NMR spectra of His-NT2RepCT hydrogels showed the presence of mainly α-helical and β-sheet, and to a lesser extent random coil conformations (Fig. 4b). Analysis of the chemical shifts of methionine residues, which are present only in NT, suggested that this domain is converted into β-sheet structure. Time-dependent spectra of NT in solution indicated a uniform reduction of signal intensities (Fig. 4c) and solid-state NMR of NT hydrogels showed that the majority of the NT residues had converted into β-sheet structures (Fig. 4d). The conformation of 2Rep in isolation could not be determined due to its aggregation-prone behavior. However, the solid-state NMR spectra of NTCT and His-NT2RepCT hydrogels appeared very similar (Fig. 4b; Supplementary Fig. 6b) which suggests that 2Rep contributes little to the structured parts of His-NT2RepCT hydrogels. For CT hydrogels, α-helix, β-sheet and random coil secondary structures were found to be present (Supplementary Fig. 6d). This suggests that some parts of CT remain α-helical while others convert into β-sheets. The results from NMR spectroscopy thus indicate that NT is important for hydrogel formation and convert into β-sheet conformation also when in fusion to 2Rep and CT. In line with this, we recently found potential for amyloid steric zipper formation in all 5 helices of the NT domain[45] and the Waltz algorithm predicts an amyloidogenic region in helix 1 (Fig. 4e).

## NT gelation is dependent on structural flexibility but not on dimerization

At pH <6.5 the NT dimerizes with concomitant stabilization against thermal or urea-induced denaturation[18]. To elucidate how dimerization and stability of NT affect gelation, solutions containing 100 mg/ml NT were monitored by vial inversion tests at pH 8, 7, and 6, respectively. The NT samples incubated at pH 8 and 7 both gelled after 30 min at 37 °C, but the pH 8 gel remained clear while the pH 7 gel showed visible precipitates (Fig. 5a). The solution containing NT at pH 6, in contrast, did not form a gel and large amounts of precipitates became visible after 20 min at 37 °C. This suggests that the dimers as such and/or their higher stability compared to the monomer preclude gelation. The formation of precipitates of NT at pH 7 and 6 was not expected as the NT has been reported to be soluble at 200 mg/ml[27], to readily refold after thermal denaturation and to remain α-helical also at lower pHs[18]. A plausible explanation for these discrepancies is that previously reported experiments have been performed at room temperature or lower, or at relatively low protein concentrations[16,18,19].

Physiological salt concentrations screen electrostatic interactions between NT subunits and shift dimerization of NT to lower pHs[18]. We found that the presence of 154 mM NaCl and NaF, respectively, indeed inhibits gelation (Fig. 5a, b; Supplementary Fig. 2b) and that these salts increase the thermal stability of the NT monomer (Fig. 5b, Supplementary Fig. 8) which further suggests that increased stability rather than dimerization prevents gel formation.

To further investigate the role of dimerization and protein stability in gelation, we used two mutants, NT* and NT[A72R], both of which stay monomeric also at low pH[28,30]. NT* is a charge-reversed double mutant in which the pronounced dipolar charge distribution of the monomer is leveled out thereby preventing dimerization and significantly increasing the stability of the monomer. NT[A72R] is a charge-dipole but the Ala replaced by an Arg is located in the dimer interface and the mutation thus interferes with subunit interactions required for dimerization. When incubated at 37 °C, NT* did not form a hydrogel while NT[A72R] formed an opaque gel within 15 min (Fig. 5c). Since both NT* and NT[A72R] are unable to dimerize but differ in stability of the monomer (Fig. 5d), these results strongly suggest that NT gelation is prevented by high thermodynamic stability. This is further supported by the finding that NT* forms gels when destabilized by elevated temperatures (after 8 min at 60 °C; Fig. 5c). Previously, it has been shown that the high methionine content in NT fluidizes its native fold and that six Met to Leu replacements (here referred to as His-NT-L6) strongly stabilize the NT monomer[46]. In line with the supposition that NT gel formation requires structural flexibility, we found that the stabilized His-NT-L6 mutant does not form gels at 37 °C (Fig. 5c, d). However, when incubated at 60 °C for 60 min, also His-NT-L6 forms a gel (Fig. 5c).

The ability of NT to convert into β-sheet structures and to form hydrogels seem to apply to some but not all spidroin NT domains. An NT from a different silk type and spider species, flagelliform silk from *Trichonephila clavipes* (NT[FlSp]), forms gels despite having a relatively low Met content and high thermal stability (Fig. 5e, f and Supplementary Table 2). In contrast, an NT from the minor ampullate spidroin from *Araneus ventricosus* (NT[MiSp]) with low thermal stability and high methionine content did not form hydrogels (Supplementary Table 2 and Fig. 5e, f). The latter finding could be due to the presence of an intramolecular disulfide bond[29,47]. In line with this, when the disulfide bond of NT[MiSp] was reduced it formed hydrogels after 10 minutes upon incubation at 37 °C (Fig. 5e). In summary, structural flexibility seems important but not the sole denominator for NT gel formation. Another factor that could be of importance is the propensity to form amyloid fibrils, and analyses by the zipper database and Waltz algorithm indeed suggest a correlation between the ability to form gels and presence of amyloidogenic regions as well as extent of regions predicted to form steric zippers (Supplementary Table 2 and Supplementary Fig. 9).

## NT fusion proteins form hydrogels in which the fusion partner remains active

NT's ability to fibrillate and form gels under benign conditions led us to hypothesize that NT fused to other protein moieties still can form gels with intact function of the fusion partner. To test this, we introduced Green fluorescent protein (GFP) and the enzyme purine nucleoside phosphorylase (PNP), respectively, C-terminally of NT. The resulting fusion proteins were expressed in *E. coli* at very high final yields (150 mg/l and 256 mg/l shake flask culture for His-NT-GFP and His-NT-PNP, respectively) in line with what has been shown for other proteins fused to NT Ref. [30]. The His-NT-GFP (300 mg/ml) and His-NT-PNP (100 mg/ml) fusion proteins formed gels after 2 and 6.5 h at 37 °C, respectively, and importantly, the GFP moiety remained folded also after gelation as >70% of the initial fluorescence intensity remains after gelation (Fig. 6a). To measure the activity of PNP in his-NT-PNP solutions and gels, we had to dilute the fusion protein with NT as the

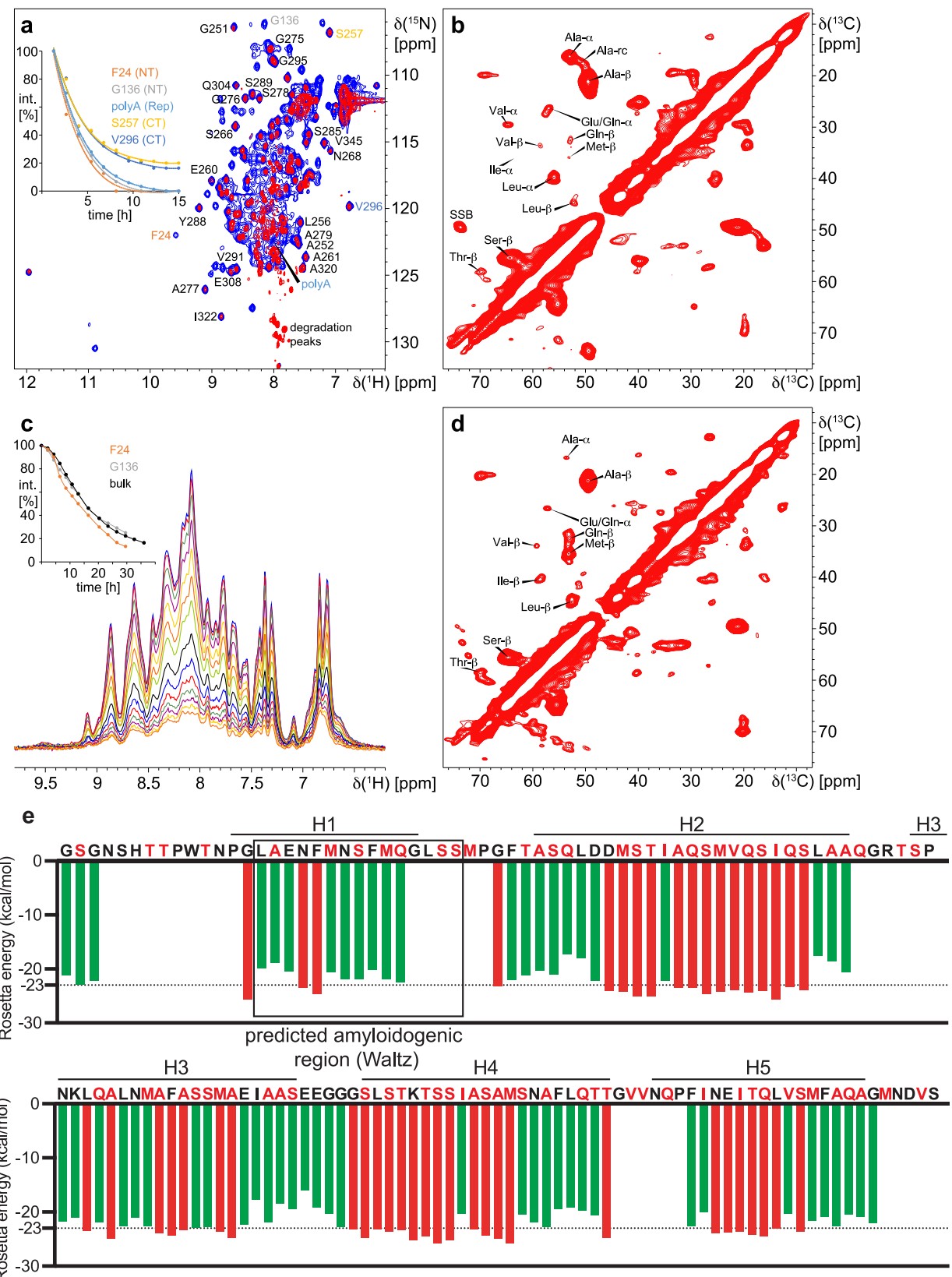

enzymatic activity of a pure preparation at concentrations relevant for gel formation was outside the detection range of the assay. Mixtures containing 0.01 mg/ml His-NT-PNP and 100 mg/ml NT formed gels that retained 65% of the initial enzymatic activity of preincubated sample (Fig. 6b). Gels remained intact during the measurements (Supplementary Fig. 10).

## Discussion

Herein, we report the formation of hydrogels from NT and other recombinant spidroins simply by incubation of the protein solutions at 37 °C (Fig. 1). We show that gelation is associated with α-helix to β-sheet conversion and formation of amyloid-like fibrils (Figs. 3 and 4). This finding is surprising since NT is a folded globular five-helix bundle

**Fig. 4 | NMR spectroscopy of His-NT2RepCT and NT and fibrillation propensity profile of NT. a** 2D ¹⁵N-HSQC spectra of a 10 mg/ml His-NT2RepCT solution before (blue) and after (red) incubation at 37 °C for 19 h. Isolated cross-peaks in the red spectrum and F24, G136, polyA in the blue spectrum are assigned using one-letter amino acid symbols and residue numbers. The inset shows signal intensity versus time for selected residues from the NT, 2Rep and CT domains. **b** Solid-state radio-frequency driven recoupling (RFDR) spectra of His-NT2RepCT hydrogels. Cα/Cβ correlations of residues observed in the RFDR spectrum determined by comparison with the chemical shifts of model peptides and values obtained from statistical data[82,83] and their secondary structure are indicated. SSB – spinning side band. **c** 1D ¹⁵N-HSQC spectra of a 10 mg/ml NT solution during incubation at 37 °C for 36 h. The inset shows bulk intensity versus time. **d** Solid-state RFDR spectra of NT hydrogels. Cα/Cβ correlations of the residues observed in the RFDR spectrum and their

secondary structure are indicated. **e** Fibrillation propensity profile of NT[45,79] according to the Zipper database (https://services.mbi.ucla.edu/zipperdb/). Rosetta energies in kcal/mol of moving windows of hexapeptide steric zippers are shown. Red bars indicate hexapeptides with high fibrillation propensities (Rosetta energies below −23 kcal/mol; below dotted line). Green bars indicate Rosetta energies above the threshold which hence are segments that are unlikely to form steric zippers. Segments containing proline are omitted from the analysis (no bars). The square indicates a predicted amyloidogenic region according to the Waltz algorithm[81], (https://waltz.switchlab.org). Amino acid residue sequence of NT on top with residue types found in β secondary structure (as determined by solid-state NMR spectroscopy) indicated in red. The positions of the five α-helices of NT are indicated (H1-H5)[28].

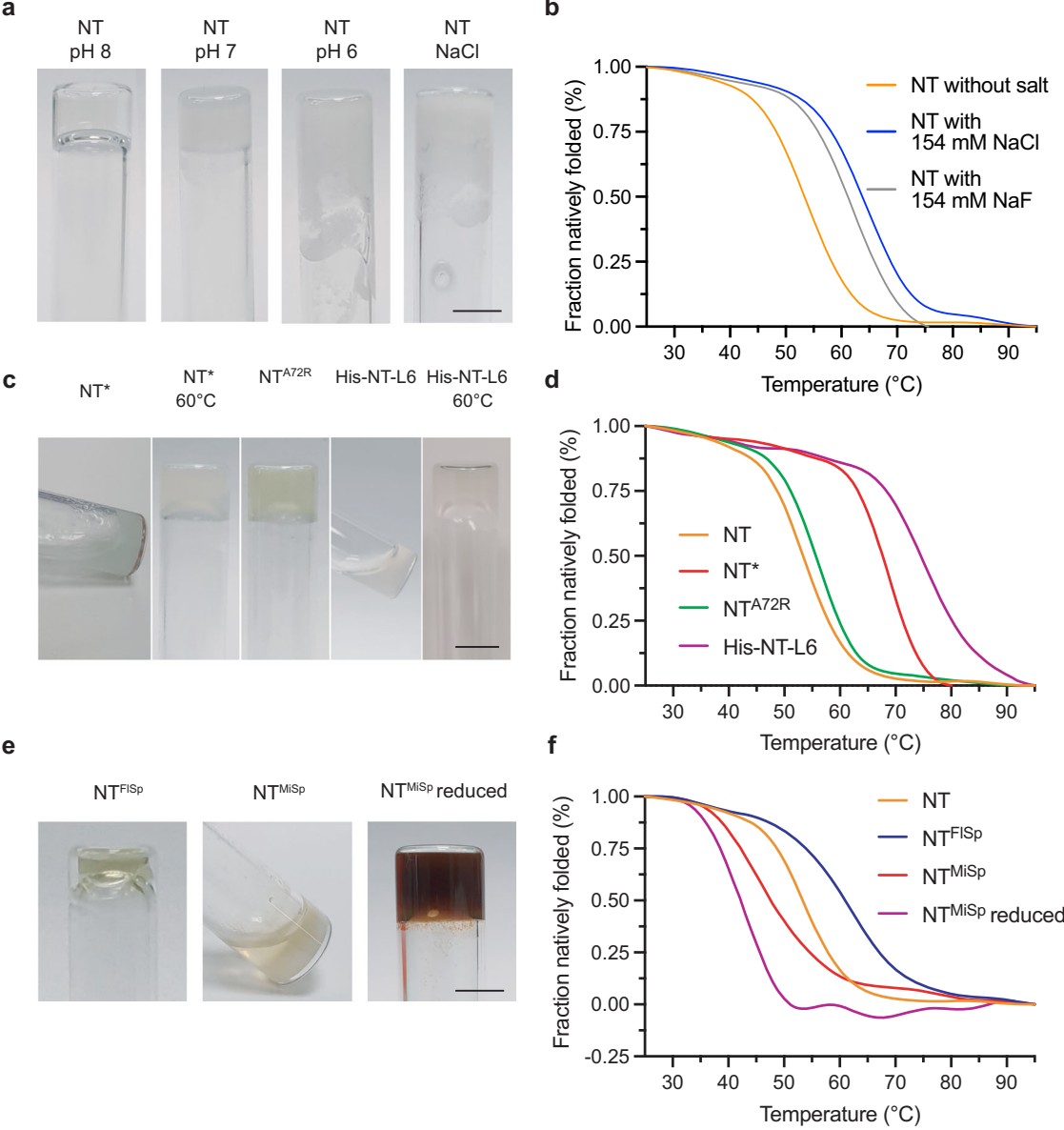

**Fig. 5 | Hydrogel formation of different NTs and their thermal stability. a** Vial inversion tests of NT (100 mg/ml) at pH 8, 7, 6 and with 154 mM NaCl (pH 8) after incubation at 37 °C. **b** CD spectroscopy of NT with and without 154 mM NaF and 154 mM NaCl, respectively. Molar ellipticity at 222 nm converted to fraction natively folded. **c** Vial inversion test of NT mutants (100 mg/ml) NT* (37 °C and 60 °C) NT^A72R (37 °C) and His-NT-L6 (37 °C and 60 °C). **d** CD

spectroscopy of NT mutants NT*, NT^A72R and His-NT-L6. Molar ellipticity at 222 nm converted to fraction natively folded. **e** Inversion test of NT^FlSp, NT^MiSp and reduced NT^MiSp (100 mg/ml). Scale bars are 5 mm. **f** CD spectroscopy of NT, NT^FlSp, NT^MiSp and reduced NT^MiSp. Molar ellipticity at 222 nm converted to fraction natively folded. Full spectra of the NTs at 25 °C and 95 °C is shown in Supplementary Fig. 8.

a

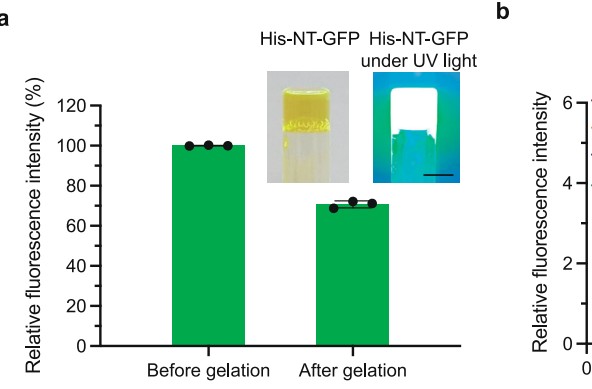

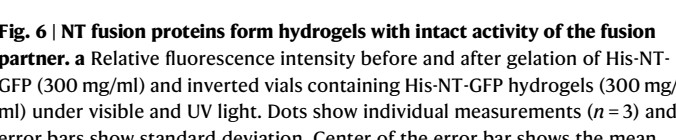

b

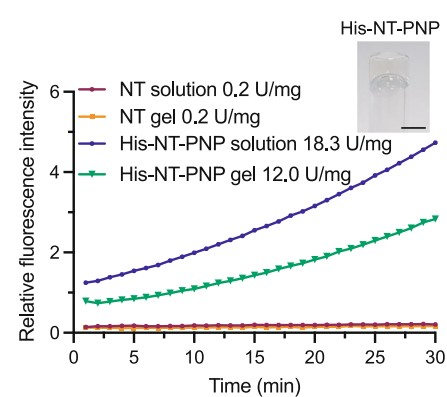

**Fig. 6 | NT fusion proteins form hydrogels with intact activity of the fusion partner. a** Relative fluorescence intensity before and after gelation of His-NT-GFP (300 mg/ml) and inverted vials containing His-NT-GFP hydrogels (300 mg/ml) under visible and UV light. Dots show individual measurements (*n* = 3) and error bars show standard deviation. Center of the error bar shows the mean.

**b** PNP activity obtained through a fluorometric assay using a solution and gel, respectively, composed of NT (100 mg/ml), and a solution and a gel formed from mixtures containing 0.01 mg/ml his-NT-PNP and 100 mg/ml NT. Insert shows an inverted vial with a His-NT-PNP containing hydrogel (scale bars are 5 mm).

that is known for its extreme solubility and high stability at concentrations of >200 mg/ml at 4 °C for days[27]. Furthermore, NT readily refolds after thermal denaturation at low μM protein concentrations[16,18,27]. According to our results, a combination of a protein concentration of >10 mg/ml and slightly increased temperatures are necessary for fibril formation (Fig. 1). This is in line with the notion that amyloid fibrils can form from globular folded proteins that expose locally unfolded states via thermal fluctuations under physiological conditions[48]. Examples of proteins that undergo such conversion include insulin[49,50], $\beta_2$-microglobulin, transthyretin and lysozyme[51–53]. Although NT in its native state is α-helical, circa 65% of the polypeptide chain is compatible with steric zipper formation (Fig. 4e)[45]. Since the monomer is dynamically mobile[46] it could expose these potentially amyloidogenic regions at moderately increased temperatures and, at high total protein concentrations, may reach critical concentrations for amyloid fibril formation[54]. In line with this reasoning, we found that spidroin concentration is negatively correlated to gelation time (Fig. 1c), and that if the conformation of the monomer NT is stabilized, i.e., by mutations (NT*, His-NT-L6) or by addition of salt, hydrogel formation is prevented (Fig. 5).

Amyloid fibrils in most cases go out of solution as precipitates, but under certain conditions they can form hydrogels[55–57]. The fibrils that form hydrogels usually have a high aspect ratio and form a stable three-dimensional network through molecular entanglement[55,58], which is in accordance with our results. For in vitro hydrogel formation, proteins are usually fully or partially unfolded, for example by organic solvents, high temperatures (70–90 °C) and/or low pH (1.5–3.0)[59–62]. For the spidroin hydrogels reported here, no harsh treatment is needed, and no cross-linking agents are required for stabilization of the hydrogels.

Spidroin repetitive segments and the CT, that apparently undergo β-sheet conversion during silk spinning, have been previously reported to form hydrogels. Compared to our findings, incubation times and/or incubation temperatures have then been significantly longer or higher, respectively, and the hydrogels formed are often opaque (Fig. 7 and Supplementary Table 1)[37,38,63–69]. In addition to the fast gelation time, NT hydrogels at > 300 mg/ml (30%) outperform all other reported recombinant spider silk protein hydrogels as well as natural hydrogels such as gelatin, alginate (2%), agar (0.5%) and collagen (0.6%) in terms of mechanical properties (Fig. 7 and Supplementary Tables 1 and 3)[37,39,66–74].

Spiders apparently have developed means to prevent gelation during spidroin storage. Although the protein concentration in the silk gland is high, the repetitive region is large in relation to the terminal

domains, meaning that the apparent concentrations of NT and CT in the gland correspond to around 10–20 mg/ml which is at the border for what is needed for the in vitro hydrogel formation observed in this study. Furthermore, salt concentrations similar those in the silk gland[16] stabilize NT (Fig. 5b). The conformation of NT has been studied in the cytosol of *E. coli* bacteria and was found to adopt a tighter fold than when studied in vitro[45], again suggesting that salts or other factors keep it from aggregating in vivo. Still, the ability of NT to convert into β-sheet fibrils may be important for silk formation and should be investigated in future studies.

In addition to the novel aspects of NT amyloid-like fibril and hydrogel formation observed in this study, we show that this phenomenon may be possible to employ for biotechnological and biomedical applications (Fig. 8). As a proof of concept, we fused NT to GFP or PNP and showed that also the fusion proteins form hydrogels when incubated at 37 °C, and that the GFP and PNP moieties maintain their activity after gelation to a large extent (Fig. 6). Nucleoside phosphorylases are valuable catalysts for synthesis of nucleoside analogues[75] which makes our findings relevant for biopharmaceutical industries. The concept of expressing fusion proteins that form transparent hydrogels under benign conditions offers the possibility of making functionalized hydrogels with advantageous properties for a wide range of applications such as immobilization of enzymes, controlled drug release and tissue engineering. Additionally, NT and NT* are efficient expression tags[30] which means that NT and variants thereof can be harnessed for high-yield production of soluble fusion proteins and subsequent generation of immobilized target proteins in 3D hydrogels.

## Methods
### Protein expression and purification
The constructs (complete list including amino acid sequences are found in Supplementary Table 4) were cloned into a pT7 plasmid and transformed into BL21 (DE3) *Escherichia coli*. Luria broth supplemented with kanamycin (70 mg/l) was inoculated with *E. coli* containing the construct plasmids and grown overnight at 30 °C and 250 rpm. The culture was then 1/100 inoculated in LB medium containing kanamycin and cultured at 30 °C and 110 rpm until OD$_{600}$ reached 0.8. For NMR studies, the bacteria were grown in M9 minimal medium containing 2 g D-glucose $^{13}$C (Aldrich) and 1 g ammonium chloride $^{15}$N (Cambridge Isotope Laboratories, Inc.) to isotope label the proteins. The temperature was lowered to 20 °C and protein expression was induced by 0.15 mM isopropylthiogalactoside (final concentration). After overnight protein expression, cells were harvested at 7278 x g,

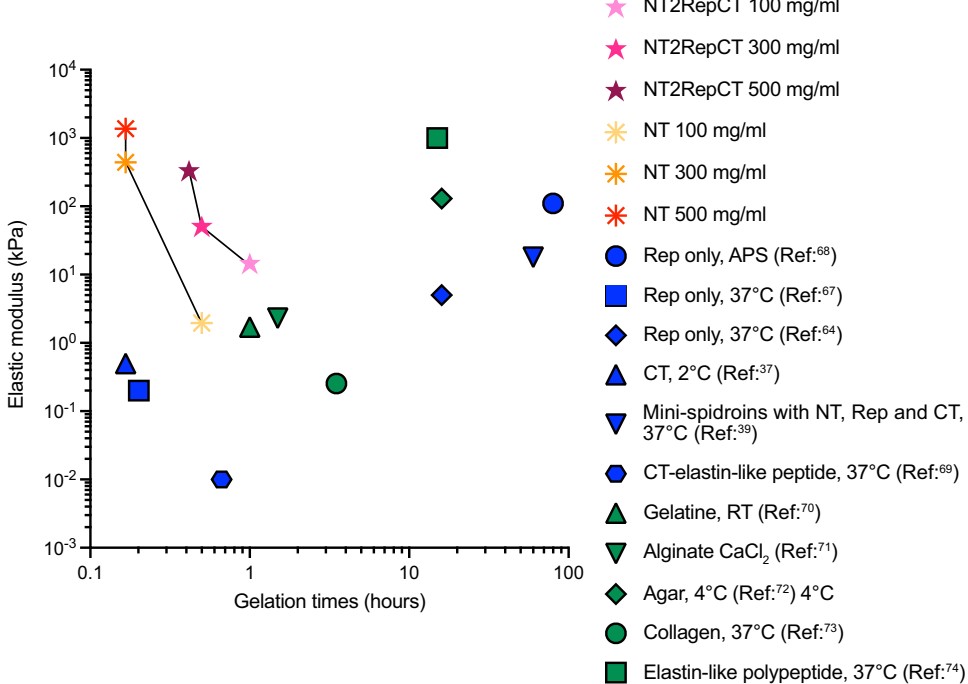

Legend:
- ★ NT2RepCT 100 mg/ml
- ★ NT2RepCT 300 mg/ml
- ★ NT2RepCT 500 mg/ml
- ✳ NT 100 mg/ml
- ✳ NT 300 mg/ml
- ✳ NT 500 mg/ml
- ● Rep only, APS (Ref:[68])
- ■ Rep only, 37°C (Ref:[67])
- ◆ Rep only, 37°C (Ref:[64])
- ▲ CT, 2°C (Ref:[37])
- ▼ Mini-spidroins with NT, Rep and CT, 37°C (Ref:[39])
- ⬡ CT-elastin-like peptide, 37°C (Ref:[69])
- ▲ Gelatine, RT (Ref:[70])
- ▼ Alginate CaCl$_2$ (Ref:[71])
- ◆ Agar, 4°C (Ref:[72]) 4°C
- ● Collagen, 37°C (Ref:[73])
- ■ Elastin-like polypeptide, 37°C (Ref:[74])

**Fig. 7 | Hydrogel properties.** Gelation time and storage modulus of hydrogels from the present study in comparison to other spidroin-based hydrogels and selected natural hydrogels. Literature references are given together with an indication of the conditions under which the gels formed. APS Ammonium persulfate; RT Room temperature. Data from[37–39,64–74].

4 °C for 20 min. Cell pellets were resuspended in 20 mM Tris-HCl pH 8 and frozen until further use. Thawed cells were lysed with a cell disrupter (T-S Series Machine, Constant Systems Limited, England) at 30 kPsi. The lysate was then centrifuged at 25,000 x g, 4 °C for 30 min. For NT$^{MiSp}$, the pellet was then resuspended in 2 M urea 20 mM Tris-HCl pH 8 and sonicated for 2 min (2 s on/off, 65%) and centrifuged again at 25,000 x g, 4 °C for 30 min. Supernatants were loaded on Ni-NTA columns, washed with 20 mM Tris-HCl, 2 mM imidazole pH 8 and finally the protein was eluted with 20 mM Tris-HCl, 200 mM imidazole pH 8. To produce NT2RepCT and NTCT, a thrombin cleavage site (ThrCleav) was introduced between His and NT. Thrombin cleavage sites were also present in His-NT-ThrCleav-2Rep (to produce 2Rep), His-Thioredoxin-ThrCleav-NT (to produce NT), His-Thioredoxin-ThrCleav-CT (to produce CT), His-Thioredoxin-ThrCleav-NT* (to produce NT*), His-Thioredoxin-ThrCleav-NT$^{A72R}$ (to produce NT$^{A72R}$), His-Thioredoxin-ThrCleav-NT$^{FlSp}$ (to produce NT$^{FlSp}$) and His-Thioredoxin-ThrCleav-NT$^{MiSp}$ (to produce NT$^{MiSp}$). Those constructs were cleaved with thrombin (1:1000) and dialyzed against 20 mM Tris-HCl pH 8, at 4 °C overnight, using a Spectra/Por dialysis membrane with a 6–8 kDa molecular weight cutoff. After dialysis, the solution was loaded on a Ni-NTA column and the flow-through containing the target proteins was collected. The protein concentrations were determined by measuring the UV absorbance at 280 nm using the extinction coefficient of the respective proteins except for NT$^{FlSp}$ where a Bradford Assay was used according to the manufacturers protocol. The purity was determined by SDS-polyacrylamide (4–20%) gel electrophoresis and Coomassie Brilliant Blue staining. The proteins were concentrated using centrifugal filter units (VivaSpin 20, GE healthcare) with a 10 kDa molecular weight cutoff at 4000 x g in rounds of 20 min.

### Gelation of proteins
Protein solutions were thawed, and 150 µl carefully pipetted in 1 ml clear sepcap vials (8 × 40 mm Thermo Scientific). Tubes were closed with caps and sealed with Parafilm to prevent evaporation. Samples (n = 3) were incubated at 37 °C or 60 °C, respectively, and periodically inverted to observe gelation. Samples that did not gel were incubated

for at least one week. The disulfide bridge of NT$^{MiSp}$ was reduced with 10 mM DTT per 10 µM protein. To analyze gelation of native spider silk dope, a Swedish bridge spider was dissected and the two major ampullate glands were put in 200 µl of 20 mM Tris-HCl buffer pH 8 and cut to allow the dope to flow out of the glands. The gland content was dissolved in the buffer of which 50 µl was used to determine the dry weight (by incubating the open vial at 60 °C until constant weight) and 150 µl was used for gelation at 37 °C.

### Rheological analyses of His-NT2RepCT and NT solutions
The gelation behavior of NT and His-NT2RepCT was characterized at different concentrations (100–500 mg/ml, n ≥ 2) by oscillatory measurements using a Discovery HR3 rheometer (TA Instruments, New Castle, DE, USA) equipped with an upper 20 mm diameter parallel plate measuring geometry/tool made of stainless steel and the gap was set to 0.5 mm. A lower stainless steel Peltier plate was used to heat the samples from 25 °C to 45 °C and back to 25 °C at 1 °C per minute. Oscillatory measurements were performed at 0.1 Hz and within the materials Linear Viscoelastic Region using a strain of 5% and 0.5% for the 100 mg/ml and 300–500 mg/ml samples, respectively. A custom-made humidity chamber was used to prevent evaporation. Data was analyzed with Prism 9.

### FTIR spectroscopy of His-NT2RepCT and NT
A Fourier-transform spectrometer (Vertex 70, Bruker, Germany) fitted with a diamond attenuated total reflection (ATR) accessory crystal (Platinum-ATR, Bruker, Germany) and a mercury cadmium telluride-detector was used to collect infrared (IR) spectra from 800 to 3900 cm$^{-1}$ at room temperature. The ATR device as well as the optical path through the spectrometer were purged with dry, filtered air before and during the experiments. Solutions (500 mg/ml to minimize water absorption peaks in the spectra) were pipetted on to the crystal while gels (500 mg/ml) were formed prior to the measurement and then transferred onto the crystal (n = 3). 1000 scans with a resolution of 2 cm$^{-1}$ and a zero-filling factor of 2 were recorded. Second derivatives were calculated with OPUS (Bruker) using a smoothing range of

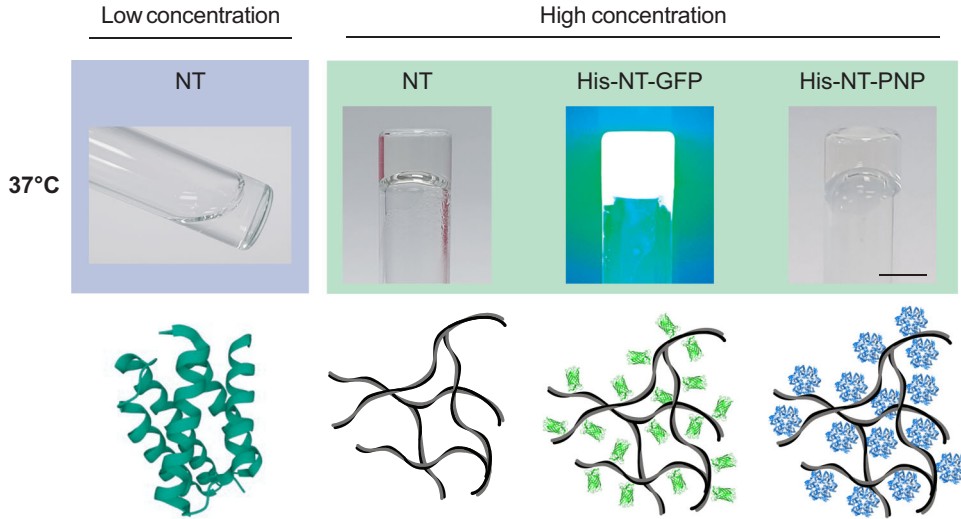

**Fig. 8 | Overview of NT gelation.** NT is soluble, α-helical and stable at low concentrations (µM) and at 37 °C. At the same temperature but at increased concentrations (>10 mg/mL), NT forms gels composed of amyloid-like fibrils. Also NT fusion proteins form fibrillar gels with intact function of the fusion moiety, which makes it possible to use NT for immobilization of various proteins in three dimensional hydrogels. Lower panel: NT (PDB: 4FBS) and illustrations of the fibrillar network and associated protein structures (hypothetical and not to scale, GFP PDB: 2B3Q, 10.2210/pdb2B3Q/pdb; PNP PDB: 4RJ2, 10.2210/pdb4RJ2/pdb).

nine points. The spectra were normalized to the same integrated area between 1720 and 1580 cm$^{-1}$ with F. Menges "Spectragryph – optical spectroscopy software". In ATR-FTIR spectroscopy, the penetration depth of the infrared beam into the sample varies with the wavenumber, causing stronger absorption at lower than at higher wavenumbers. These effects were not corrected for the spectra shown in Fig. 3 because they were found to very minor (Supplementary Fig. 4). The corrected spectra for that figure were calculated with the Bruker OPUS software.

In principle, a full quantitative analysis of protein conformations following a reliable deconvolution of components within the amide I peak may be possible. In practice, however, several obstacles exist. Noise in the spectrum can appear as (false) peaks in the deconvolution. Moreover, a peak due to the bending mode of water coincides with the amide I peak position and may be of similar magnitude for specimens containing large amounts of water (such as the aqueous gels investigated here). Consequently, we have not attempted a full deconvolution of the amide I peak, and our observations should be regarded as only providing support for results from other methods (e.g. NMR spectroscopy).

**Transmission electron microscopy (TEM)**
50 mg/ml NT and His-NT2RepCT solutions were gelled at 37 °C overnight. The hydrogels were then diluted in 20 mM Tris-HCl (pH 8) to reach a concentration of 12.5 mg/ml and thoroughly vortexed and pipetted up and down to disrupt the gels. Hydrogels were further diluted 10 times in 20 mM Tris-HCl (pH 8) and 5 µl of the samples were dropped onto a formvar coated copper grid and excess sample was removed with blotting paper. The samples were washed twice with 5 µl MilliQ water and then stained with 1% uranyl formate for 5 minutes. Excess stain was removed with blotting paper followed by air drying of the grid. Imaging was performed with an FEI Tecnai 12 Spirit BioTWIN (operated at 100 kV) on these grids. Images were recorded using a 2 k × 2 k Veleta CCD camera (Olympus Soft Imaging Solutions, GmbH, Münster, Germany) at x 26,500 and x 43,000. For each sample (*n* = 1), 10–15 images were recorded. ImageJ (https://imagej.nih.gov/) was used for image analysis and measuring of the fibril diameter (*n* = 100, different fibrils). Prism 9 was used to perform an unpaired *t*-test (two-tailed). Means were, 11.43 (SD 2.035) and 7.67 (SD 1.389) nm for His-

NT2RepCT and NT fibrils, respectively. Confidence interval (95%) was −4.246 to −3.275. Degrees of freedom = 198 and *p* < 0.0001.

**Turbidity and Thioflavin T (ThT) assays**
Corning 96 well-black clear bottom plates (Corning Glass 3881, USA) were used to measure 80 µl of the liquid samples with 10 µM Thioflavin T (ThT) in triplicate (*n* = 3) under quiescent conditions. A 440 nm excitation filter and a 480 nm emission filter were used to record the difference in fluorescence (FLUOStar Galaxy from BMG Labtech, Offenburg, Germany). The ThT signal was neither saturated nor quenched as experiments with various ThT concentrations were performed without changes in the signal intensity. The absorbance at 360 nm was recorded to measure turbidity. For seeding experiments, a 100 mg/ml gel was formed at 37 °C, resuspended and then used for seeding at 5, 10, and 20% molar ratio. Data was analyzed with Prism 9.

**Microscopy**
His-NT2RepCT and NT stocks of > 100 mg/ml were thawed on ice and filtered through 0.22 µm filters. The concentration was calculated by measuring the absorbance at 280 nm, using a Nanodrop. The samples were diluted to 20 mg/ml in 20 mM Tris-HCl pH 8 in a well of a black non-binding surface 96-well plate with clear bottom (Corning) and mixed with 5 µM ThT (final concentration) in a total sample volume of 50 µl. Samples were imaged every 10 min at 37 °C on a CellObserver microscope (Zeiss) with the transmitted light channel and with FITC excitation and emission filter sets for ThT imaging. 20x/0.4 objective lens was used for imaging. Zen Blue (Zeiss) and ImageJ (https://imagej.nih.gov/) was used for image analysis. Gels were also made from solutions of NT and His-NT2RepCT at 50 mg/ml in 20 mM Tris pH 8 with 5 µM ThT by incubation at 37 °C for 90 min. Pieces of the gels were transferred to a new well containing 20 mM Tris pH 8 and 5 µM ThT in a non-binding black 96-well plate with clear bottom. Green fluorescence and brightfield images were acquired at 20x/0.4. ImageJ was used for image analysis.

**NMR spectroscopy**
Solution NMR spectra were acquired at 310 K on a 600 MHz Bruker Avance Neo spectrometer equipped with a QCI quadruple-resonance (HFCN) pulsed-field-gradient cryoprobe. The NMR samples contained

10 mg/ml uniformly $^{13}C$, $^{15}N$ labeled protein dissolved in 20 mM Tris-HCl (pH 8), 0.02% (w/v) $NaN_3$, 5% $D_2O$ (v/v), ($n = 1$). The chemical shifts of NT2RepCT at pH 6.7 were used for assignment of peaks in the 2D $^{15}N$-HSQC spectra[23].

Solid-state magic-angle spinning (MAS) NMR spectra for $^{13}C$, $^{15}N$ labeled hydrogels were recorded on an 800 MHz Bruker Avance III HD spectrometer equipped with a 3.2 mm $^{13}C/^{15}N\{^{1}H\}$ E-free probe. The temperature of the sample was regulated using variable-temperature gas flow at 277 K. 2D dipolar-assisted rotational resonance (DARR)[76], and radio-frequency driven recoupling (RFDR)[77], spectra were obtained at a MAS frequency of 12.5 kHz and 20 kHz, respectively. Cross-polarization (CP) from $^{1}H$ to $^{13}C$ was accomplished using a linear ramp from 60.0 to 48.0 kHz on $^{1}H$, 61.3/71.6 kHz on $^{13}C$ (at 12.5/20 kHz MAS) and a contact time of 0.5–1 ms. Spinal64[78] decoupling at 73.5 kHz amplitude was applied during acquisition. The acquisition time was 10 ms and the recycle delay was 2.5 s. One-bond $C\alpha/C\beta$ correlations observed in the RFDR spectra were assigned based on residue type characteristic chemical shifts and multiple-bond correlations in the DARR spectrum.

### Fibrillation propensity and amyloidogenic regions prediction in NT

The Zipper database[79], (https://services.mbi.ucla.edu/zipperdb/) was used to estimate the fibrillation propensity and Rosetta energies of NT, $NT^{FlSp}$ and $NT^{MiSp}$. The Zipper database calculates the Rosetta energy[80] which combines several free energy functions to model and analyze protein structures. Energies equal to or below −23 kcal/mol indicate high fibrillation propensity. Lower energies imply higher stability of two β-strands in a zipper conformation. Additionally, the Waltz algorithm was used to predict amyloidogenic regions in NT, $NT^{FlSp}$ and $NT^{MiSp}$ Ref. [81]. (https://waltz.switchlab.org/).

### Preparation of low pH protein solutions

NT protein solutions were mixed with 2-(N-morpholino)ethanesulfonic acid (MES) buffer at pH 5.5 and 6.0 to lower the pH to pH 6 and 7, respectively. The final protein concentration was 100 mg/ml.

### Circular dichroism spectroscopy

Measurements were performed on a J-1500 CD spectrometer (JASCO, USA) using a 300 μl cuvette with a 0.1 cm path length. Proteins were diluted to 10 μM in 20 mM phosphate buffer (pH 8), ($n = 1$). To analyze the stability of the proteins in the presence of salt, these were analyzed at the same concentration in 20 mM phosphate buffer (pH 8) with 154 mM NaF or NaCl, respectively ($n = 1$). Temperature scans were recorded at 222 nm between 25 °C to 95 °C at a heating rate of 1 °C min$^{-1}$. The fraction of natively folded protein was calculated with the formula ($CD_{measured} − CD_{final}$)/($CD_{beginning} − CD_{final}$). Furthermore, five spectra of each sample were recorded from 260 nm to 190 nm at 25 °C and after heating to 95 °C. The five spectra were averaged, smoothed and converted to molar ellipticity. Data was analyzed with Prism 9.

### GFP activity assay

The fluorescence intensity of His-NT-GFP (300 mg/ml, 80 μl) was measured in a Corning 96 well-black clear bottom plate (Corning Glass 3881, USA) in triplicate ($n = 3$) under quiescent conditions. The samples were measured with a fluorescence-based plate reader using an excitation wavelength of 395 nm and emission was recorded at 509 nm before gelation and after 2 h at 37 °C. Data were analyzed with Prism 9.

### PNP activity assay

Purine Nucleoside Phosphorylase Activity Assay Kit (Fluorometric, Sigma Aldrich) was used according to the manufacturer's instructions. For activity measurements in the gels and solutions containing His-NT-PNP, 10 ng of His-NT-PNP was mixed with 100 mg/ml NT to a total volume of 2 μl as gels made from pure His-NT-PNP resulted in signals above the detection interval of the kit. Controls of gels and solutions without His-NT-PNP were included. Measurements were performed in duplicates ($n = 2$). After the activity measurements, the reaction mix was removed and the gels were photographed to ensure that the gels remained intact during the measurements. Data was analyzed with Prism 9.

### Reporting summary

Further information on research design is available in the Nature Research Reporting Summary linked to this article.

## Data availability

Source data are provided for Figs. 1c, 2a–c, 3a, b, e–g, 4, 5 b, d, f, and 6, Supplementary Fig. 3, Supplementary Fig. 5a, d, Supplementary Fig. 6 and Supplementary Fig. 8. The data generated in this study have been deposited in the Zenodo database https://doi.org/10.5281/zenodo.6683653. The NMR data generated in this study have been deposited in the BMRBig repository with entry ID bmrbig36. GFP and PNP structures were taken from the PDB (GFP 2B3Q, PNP 4RJ2).

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

## Acknowledgements

This work was supported by European Research Council (ERC) under the European Union's Horizon 2020 research and innovation program (grant agreement No 815357), the Center for Innovative Medicine (CIMED) at Karolinska Institutet and Stockholm City Council, Karolinska Institutet SFO Regen (FOR 4-1364/2019), and the Swedish Research Council (2019-01257) to A.R. K.J. acknowledges support from European Regional Development Fund project No. 1.1.1.1/18/A/004. C.S. is supported by the Novo Nordisk Foundation (Grant no.: NNF19OC0055700). We acknowledge support from Florian Salomons at the BIC core facility at Karolinska Institutet.

## Author contributions

T.A., A.R., and J.J. conceived and designed the study. K.J. performed N.M.R. spectroscopy. T.A., J.F., O.S. performed experimental work related to cloning, protein expression, and purification. T.A. and U.C. performed vial inversion experiments. T.A. and J.F. performed ThT and turbidity experiments. R.S. contributed with ideas related to use of PNP as a model enzyme. T.A. and O.S. performed PNP activity assay. M.J., P.R.L., E.B.L., and T.A. performed rheological characterization. P.R.L., E.B.L., and T.A. performed FTIR spectroscopy. U.C. performed silk gland dissection. R.K. and G.C. performed TEM. C.S. performed microscopy. T.A. and U.C. performed C.D. spectroscopy measurements. A.A., N.K., M.Langton, C.H., M. Landreh, A.B., J.J., and A.R. contributed by supervision and data analyses. T.A. wrote the main part of the manuscript with contribution from A.R. and J.J. All authors have read and commented on the manuscript.

## Funding

## Competing interests
