## [Peer Review File · Nature Communications]

Spidroin N-terminal domain forms amyloid-like fibril based hydrogels and provides a protein immobilization platformREVIEWER COMMENTS

Reviewer #1 (Remarks to the Author):

Arndt et al. present an exciting and quite comprehensive demonstration of hydrogel formation by recombinant spider silk proteins. As is clearly described and shown, this behaviour contrasts with prior demonstrations of hydrogel formation by spidroins in that it occurs quite rapidly (i.e., within minutes) and with only a modest elevation in temperature (37 °C) rather than requiring either a length incubation period, elevated temperature, and/or chemical initiation. Very intriguingly, the contributions of the components of the recombinant spidroin are “dissected”, with the gelation behaviour being shown to be tied to the non-repetitive N-terminal domain (“NT” in the parlance of the manuscript). This spidroin domain is shown to be sufficient for gelation on its own, with a direct tie to β -sheet formation shown both through FTIR and through heteronuclear NMR spectroscopy techniques. Amyloid fibril-like self-assembly behaviour is argued to be occurring during gelation, as supported by thioflavin T assays and electron microscopy analysis with direct coupling to computational analysis. The effects of NT stability/dynamics vs. dimerization are then investigated, with dimerization shown to not be required whereas increased NT stability (even in the monomeric context) reduces gelation propensity. Finally, the ability of NT to form hydrogels when fused N-terminally to another protein of interest is shown. Two distinct proteins are shown to remain in their functional form upon hydrogel formation – green fluorescent protein (GFP) and purine nucleoside phosphorylase (PNP). As a whole, this is a very well-written manuscript, with most experimental results, analyses and conclusions being robustly detailed and interpreted. In my opinion, this work will be of wide interest, and the demonstration that NT acts to promote hydrogel formation while allowing disparate fused proteins to retain their folded/functional state makes this highly compelling in terms of potential applicability beyond spidroins.

Notwithstanding my general enthusiasm and the generally excellent state of the manuscript as it stands, there are several points that I feel would merit further consideration and/or clarification:

1) Is it reasonable to refer to the fibrils that are produced as amyloid (or amyloid-like) when there is no evidence for seed-induced rate enhancement? Certainly, these bind ThT – consistent with a β -sheet structuring in these fibrils. But, I would be concerned that it may not be merited to go beyond that and explicitly attribute these as being amyloid(-like) without the demonstrated ability to seed and enhance fibril formation.

2) Can the authors further rationalize their observations by solution-state NMR spectroscopy (e.g., Fig. 4A)? Namely, the resonances that are still being observed in the post-incubation sample actually appear to be narrower and better resolved than those in the pre-gelation sample. This seems contrary to the idea that those resonances are associated with a protein that has become part of a hydrogel, where presumably the mobility and tumbling of the protein would be attenuated (i.e., broadening lines) rather than narrowed. Even if the CT, as these resonances are attributed to, is structurally unperturbed upon gelation, it should be hindered in motion through incorporation into the hydrogel. Any residual protein in solution, even if “non-gelled”, would also presumably be experiencing slower motion/diffusion due to the increased viscosity in the gellated sample. In short, I don’t disbelieve that the assigned resonances show that there are CT signals that appear unchanged, but I would like to see some rationalization and discussion as to why these are being observed and why they appear narrower.

3) The solid-state NMR data are quite compelling and show the structural transition very nicely. I am a little concerned, however, about the attribution of random coil character that is made in Supp. Fig. 5d – i.e., many of the correlations that are noted to be indicative of an rc-type conformation are also observable in Fig. 4b. Why are these being considered so important, relatively speaking, in analysis of Supp. Fig. 5d (CT alone) but not mentioned in Fig. 4b? I would also note that there does not appear to be a reference (or references) provided with respect to which chemical shift reference datasets are being used to make these structural inferences. Finally, I am also wondering if there are any domain-specific attributions that can be made? (i.e., without my having looked in detail at the sequences that are very nicely provided by the authors, are there certain correlations in the DARR spectrum that are specific to one domain or another

based upon amino acid compositions?) If so, this type of analysis would be highly valuable to include.

4) On p. 14, NaF is evaluated with respect to thermal stability of the NT monomer whereas NaCl is used to evaluate the effect of salt upon gelation. I am not clear as to why two different salts were employed, since these seem to be discussed as being equivalent. It would certainly be ideal to use the same salt(s) for both of these assays rather than convoluting the argument by swapping one anion for another as this just leads to questions about robustness of the finding.

5) The argument on p. 14 that elevated (NT monomer) stability is preventing formation of the hydrogel is quite elegant! To go beyond this and provide even more compelling demonstration of this behaviour, is the NT-L6 mutant also capable of gelation at elevated temperature? Given that NT* shows this behaviour, this is not by any means an essential experiment, but if these data were available that would certainly further bolster the argument.

I also have several more minor suggestions for modifications/edits - noting that there are several other minor typos/wording issues which I have not explicitly mentioned that can certainly be corrected in revision. These are detailed in order of appearance:

1) Para. spanning pp. 2-3 – the discussion about spider silk behaviour (in solution, in the gland and in the fibre) is quite heavily based on major ampullate silk, which entirely makes sense based upon our level of understanding of this silk vs. other spider silks as well as the fact that the construct being used by the authors is MaSp1 based. But, this introductory section is worded in such a way that this could be interpreted by a reader as general behaviour - I would suggest that it is best to ensure that the features and behaviour that are specific to major ampullate silk be noted as such.

2) p. 3, para. 2 – 14 g/L is indeed an impressive yield, but “impressive” probably should be reworded.

3) Para. spanning pp. 6-7 – the introductory sentence to this paragraph “Rheological measurements...” is not particularly informative, and does not provide a lot of context/lead-in for the next sentence which then introduces data. The types of moduli being measured and discussed should also be explicitly defined when these are first being reported, rather than in passing later in the paragraph. There is also some mixing of terms (storage/elastic vs. loss/viscous) in the figure captions, figures themselves, and main manuscript text (as well as later in Fig. 8/related discussion) which may overly convolute these studies. Finally, more detail about the rate of the temperature ramp may also be beneficial to include and discuss, especially with regard to its relation to the time-dependence observed for the gelation process.

4) p. 8, para. 1 – would it be more appropriate to present these results on dope extracted from major ampullate gland before the rheology data? (It seems out of place in its current context.)

5) p. 8, para. 2 (and Methods) – the FTIR data were acquired through ATR, but no mention of spectral correction for the effects of ATR is noted. Were any corrections applied for ATR – if so, this should be noted; if not, what is the rationale for not applying this correction to the data?

6) p. 10, para. 1 (and Supp. Fig. 4) – the fluorescence micrographs are not entirely convincing as they appear to be entirely uniform over the entire micrograph – i.e., wouldn't at least some minimal fluctuations in intensity and/or dependence upon density of the material being imaged be expected? These may indeed be robust, but with such a complete lack of any discernable micrograph image features it leads one to wonder if there were technical issues with the instrument during acquisition and/or processing of the data.

7) p. 10, para. 2 – I'm not convinced that the inset shown in Fig. 4c (a bulk integral over NH resonances) can be interpreted, as it has been, to show a uniform reduction of signal intensities. Instead, I would expect to see the attenuation of individual resonances analyzed and shown to be similar (within error) to demonstrate a “uniform” reduction.

8) p. 11, para. 1, line 4 – the conclusion is made that “NT is most important for hydrogel formation”. Based upon the NMR analyses, this seems like a leap – i.e., certainly there’s demonstration that this region becomes structured etc., but stating that it’s “most important” seems to go beyond the current set of data and might be more appropriate at a later point in the manuscript?

9) p. 12, Fig. 4e – with the statement in the text (p. 11) that there’s predicted amyloid character in the five helices of the NT, it would be helpful for the positions of these helices with respect to the linear sequence to also be annotated/highlighted in some way.

10) p. 14 and elsewhere – the NT^{L6} nomenclature is a little confusing, in that the same superscript style is also used to describe the single A82R mutant. I understand that the L6 mutant contains six Met -> Leu substitutions, but for ease of understanding I would suggest that a different way of referring to this mutant be used in order to ensure that it’s not thought to be referring (albeit incompletely!) to a single mutation at residue 6.

11) p. 15, panels b, d and f – are there representative CD spectra for each construct to back up these inferences about folding that are based on temperature scan measurement at only 222 nm? I would be concerned that without this context the ellipticity at 222 nm alone is not diagnostic of the “folded state” of the protein. These data certainly do not need to be in the main manuscript, but could be provided in the Supp. Info.

12) p. 16, para. 1 – ensure that species names are italicized and note that “Supplementary” Table 2 is misspelled in the last sentence. I would also suggest that the final sentence would benefit from a little more detailing about what, exactly, the correlation being proposed is.

13) p. 16, para. 2 – the second sentence would benefit from being reworded (i.e., it currently has a mix of gene- vs. protein-oriented descriptions).

14) Para. spanning pp. 16-17 and Fig 6b – it’s not fully clear that the PNP fusion is being “diluted”, nor why. The figure caption also implies that this is a sample composed entirely of PNP fusion, which is inconsistent with the text. In terms of the dilution, a reader might be left wondering if this is needed due to an issue with the enzyme remaining functional as a fusion protein if it’s undiluted. I recognize that this is mentioned in more depth in the methods section (with the argument, as I read it, that activity was “too high” to assay), but it would be valuable to detail this up front as part of the results section. I am also curious how the values of U/mg were determined – from the manner in which the data are presented, this could be interpreted as being known “a priori” but presumably this was actually determined through the fluorogenic assay?

15) p. 20, para. 1 – the arguments being made would also benefit from discussion of the anticipated temperature in the gland itself. (i.e., if a temp. of ~37 °C is needed for effective gelation, how does this compare to the temperature in the gland? Presumably that’s mostly environmentally determined for a given spider?)

16) pp. 23-26 – the protein sequences are indeed essential and the format in which they are provided is ideal, with respect to being able to be copied and pasted. However, these would likely be better provided either in a table in the main text or in the Supporting Information, rather than as a standalone segment of the Methods section.

In support of the desire by Nature Communications to increase transparency, I am happy to sign this review:

Jan K. Rainey, Ph.D.

Department of Biochemistry & Molecular Biology

Dalhousie University

Halifax, Nova Scotia, Canada

Reviewer #2 (Remarks to the Author):

The authors report the N-terminal domain (NT) and its fusion proteins form self-supporting and transparent hydrogels at 37 °C, and thus provide a protein immobilization platform. In addition, they also explore the gelation mechanism and find the gelation is caused by NT α -helix to β -sheet conversion and formation of amyloid-like fibrils. The study may be of general interest but enthusiasm is limited by a lack of novelty and concerns about the gelation mechanism.

Major concerns

1. Novelty. It is not surprising to see the proteins or peptides forming self-supporting hydrogels at 37 °C although this is the first time to be seen for NT. For example, elastin-like proteins, silk-elastin-like proteins and many other designed polypeptides could form temperature-responsive hydrogels and have been demonstrated to be used in drug delivery and tissue engineering. The gelation mechanism for these proteins and peptides is also widely explored.
2. As we know, the NTs are evolutionarily conserved within spidroins of different spider species, why do only some of them form hydrogels? Does the gelation of NT contribute to spider silk formation? This might be more interesting than gelation itself.
3. As shown in Figure 1b, the gelation of NT is time-dependent. The authors should study the evolution of transparency, secondary structures, internal morphologies of hydrogels with time, to explore the gelation mechanism.
4. As shown in Figure 2b and 2c, the gelation temperature (cross-over point when $G' > G''$) for His-NT2RepCT (300mg/ml) is around 40°C, but for NT (300mg/ml) is below 25°C. This result seems inconsistent with the findings by vial inversion in figure 1b and c. The authors should identify the gelation temperature by oscillatory rheology first and then explore the evolving of gels with time at a specific temperature.

Minor points:

5. Page 10, the authors point that "but addition of 5, 10 or 20 % (w/w) NT hydrogels to NT solutions did not give any pronounced seeding effect (Fig. 3g)." The seeds are usually nanofibrils in solutions, how could the authors use the hydrogels as seeds? The fibrils in the hydrogels are relatively fixed and not easy to be reached as seeds.
6. In figure 2, the protein concentration used for turbidity measurement is 100mg/ml, but for oscillatory measurement is 300mg/ml, why not be consistent?
7. In figure 1c and figure 2d, replicates are not included.
8. Page 16, "The His-NT-GFP (300 mg/ml) and His-NT-PNP (100 mg/ml) fusion proteins formed gels after 2 and 6.5 hours at 37 °C". From the application's point of view, the gelation concentrations are relatively high and the gelation time is long. Please discuss how to improve this for the application.
9. The storage modulus and loss modulus in figure 2b and c should be corrected to the elastic modulus and viscous modulus, to be consistent with the figure legend.
10. Ref 25 and 39 are same papers. 77 and 84 are same.

Reviewer #3 (Remarks to the Author):

This manuscript claims that recombinant N-terminal domain (NT) from spider silks can rapidly form self-supporting and transparent hydrogels at 37 °C, and also provide a protein immobilization platform that can introduce gelation when fused with other protein. Overall, most of the results support the authors' conclusions and claims. However, I recommend that the manuscript should fully address the following questions:

1. The gelation of recombinant spider silk proteins is very common in lots of other researches, so what is the significance in this article?

2. The authors mentioned that "All hydrogels, except 2RepCT gels, were optically transparent to the naked eye (Fig. 1b). This was verified for all gels by measuring light transmission, with exception of NT2Rep and CT gels which showed an increase in absorbance at 360 nm over time (Fig. 2a)."

As presented in Figure 2b, I think not all the hydrogels were optically transparent, for example, HIS-NT2RepCT, NTCT. Also, the authors claimed optically transparent, but why the authors only measured the light absorbance at 360 nm? To my knowledge, 360 nm is in the range of UV light, rather than in the visible light ranged from 380-780 nm.

3. The authors mentioned that "The band at 1617 cm^{-1} was more pronounced for NT than for His-NT2RepCT, suggesting a higher overall beta-sheet content in NT hydrogels than in NT2RepCT hydrogels."

I recommend the authors to revise this statement. First, deconvolution of the peaks is necessary if the authors want to discuss the content of beta-sheet. Then, the ATR model is not favorable for the deconvolution of the specific peaks because lights in this model do not go through the whole sample, that is not following the Beer-Lambert law. The surface transparency of the samples will significantly affect their absorbance peaks.

4. The authors mentioned that "Analysis of the gel by transmission electron microscopy (TEM) revealed that the hydrogels are composed of amyloid-like fibrils (Fig. 3c-d)."

Do the authors check the orientation of the beta-sheet in this nanofibrils? Cross-beta or not is the difference between the amyloid-like fibrils and the silk fibroin-like fibrils.

5. More details should be provided for the hydrogels of the fusion proteins. For example, how to express the proteins and to form hydrogels? Do the authors run the PAGE to check the expression? Do the authors consider the free proteins in the hydrogels and solution?

REPLY TO REVIEWER COMMENTS

We are grateful to the reviewers for their efforts and suggestions to improve our manuscript. Below is a detailed response to the questions/comments raised.

Reviewer #1 (Remarks to the Author):

Arndt et al. present an exciting and quite comprehensive demonstration of hydrogel formation by recombinant spider silk proteins. As is clearly described and shown, this behaviour contrasts with prior demonstrations of hydrogel formation by spidroins in that it occurs quite rapidly (i.e., within minutes) and with only a modest elevation in temperature (37 °C) rather than requiring either a length incubation period, elevated temperature, and/or chemical initiation. Very intriguingly, the contributions of the components of the recombinant spidroin are “dissected”, with the gelation behaviour being shown to be tied to the non-repetitive N-terminal domain (“NT” in the parlance of the manuscript). This spidroin domain is shown to be sufficient for gelation on its own, with a direct tie to β -sheet formation shown both through FTIR and through heteronuclear NMR spectroscopy techniques. Amyloid fibril-like self-assembly behaviour is argued to be occurring during gelation, as supported by thioflavin T assays and electron microscopy analysis with direct coupling to computational analysis. The effects of NT stability/dynamics vs. dimerization are then investigated, with dimerization shown to not be required whereas increased NT stability (even in the monomeric context) reduces gelation propensity. Finally, the ability of NT to form hydrogels when fused N-terminally to another protein of interest is shown. Two distinct proteins are shown to remain in their functional form upon hydrogel formation – green fluorescent protein (GFP) and purine nucleoside phosphorylase (PNP). As a whole, this is a very well-written manuscript, with most experimental results, analyses and conclusions being robustly detailed and interpreted. In my opinion, this work will be of wide interest, and the demonstration that NT acts to promote hydrogel formation while allowing disparate fused proteins to retain their folded/functional state makes this highly compelling in terms of potential applicability beyond spidroins.

Reply: We thank you Jan for your supportive assessment of our work. We really appreciate your comments below and have endeavored to address all of them with amendments to the manuscript to improve readability, accessibility and clarification of our findings.

Notwithstanding my general enthusiasm and the generally excellent state of the manuscript as it stands, there are several points that I feel would merit further consideration and/or clarification:

1) Is it reasonable to refer to the fibrils that are produced as amyloid (or amyloid-like) when there is no evidence for seed-induced rate enhancement? Certainly, these bind ThT – consistent with a β -sheet structuring in these fibrils. But, I would be concerned that it may

not be merited to go beyond that and explicitly attribute these as being amyloid(-like) without the demonstrated ability to seed and enhance fibril formation.

Reply: This is a good point. Seeding is not required for the classification as amyloid and the fibrils we observe show morphological features and ThT binding that are characteristic of amyloid fibrils. However, to ensure we use the correct nomenclature, we now refer to the fibrils as amyloid-like, and also changed the title of the manuscript to conform to this.

2) Can the authors further rationalize their observations by solution-state NMR spectroscopy (e.g., Fig. 4A)? Namely, the resonances that are still being observed in the post-incubation sample actually appear to be narrower and better resolved than those in the pre-gelation sample. This seems contrary to the idea that those resonances are associated with a protein that has become part of a hydrogel, where presumably the mobility and tumbling of the protein would be attenuated (i.e., broadening lines) rather than narrowed. Even if the CT, as these resonances are attributed to, is structurally unperturbed upon gelation, it should be hindered in motion through incorporation into the hydrogel. Any residual protein in solution, even if “non-gelled”, would also presumably be experiencing slower motion/diffusion due to the increased viscosity in the gellated sample. In short, I don’t disbelieve that the assigned resonances show that there are CT signals that appear unchanged, but I would like to see some rationalization and discussion as to why these are being observed and why they appear narrower.

Reply: The reviewer is right that the NMR signal linewidths before and after gelation are comparable or even slightly narrower in the post-incubation sample. However, as shown in the inset of Fig. 4A, the CT signals are also attenuated to ~20% of their original intensity. We attribute the intensity reduction to CTs that are fully incorporated into the hydrogel structure and thereby have become invisible to solution NMR spectroscopy, whereas the remaining intensity may be from a smaller fraction of CTs, which remain as mobile as in the pre-incubation sample. The most N-terminal CT residue that is observed by NMR in the post-incubation sample is G251, while the first ~10 structured residues are not observed, probably due to hindered motion through incorporation of their attached NT2Rep part into the hydrogel. We have added a discussion around these lines on p. 11-12.

3) The solid-state NMR data are quite compelling and show the structural transition very nicely. I am a little concerned, however, about the attribution of random coil character that is made in Supp. Fig. 5d – i.e., many of the correlations that are noted to be indicative of an rc-type conformation are also observable in Fig. 4b. Why are these being considered so important, relatively speaking, in analysis of Supp. Fig. 5d (CT alone) but not mentioned in Fig. 4b? I would also note that there does not appear to be a reference (or references) provided with respect to which chemical shift reference datasets are being used to make these structural inferences. Finally, I am also wondering if there are any domain-specific attributions that can be made? (i.e., without my having looked in detail at the sequences that are very nicely provided by the authors, are there certain correlations in the DARR spectrum that are specific to one domain or another based upon amino acid compositions?) If so, this type of analysis would be highly valuable to include.

Reply: Fig. 4b shows the solid-state NMR spectrum of His-NT2RepCT. Comparison of this spectrum with those of NT alone (Fig. 4d) and CT alone (Supp. Fig. 6d) allows to make attribution of the random coil character specifically to the CT. This goes in line with the conclusion made in the previous point, i.e., that CT is also largely incorporated into the hydrogel structure of His-NT2RepCT, and thus contributes its intensity to the spectrum of Fig. 4b. We have changed Fig. 4b and the text to indicate presence of random coil conformation in the His-NT2RepCT hydrogel. We have also added references in the caption of Fig. 4 and Supp. Fig. 6 to chemical shift reference datasets derived from model peptides and from statistical data. The only domain-specific residue types are Met/Phe/Trp (present in NT only) and Tyr (present in Rep and CT, but not in NT). Of these, only Met could be assigned in the solid-state NMR spectra and was found in β -sheet conformation, which corroborates the findings that majority of the NT residues are converted into β -sheet structures. A sentence addressing this has been added to p 12.

4) On p. 14, NaF is evaluated with respect to thermal stability of the NT monomer whereas NaCl is used to evaluate the effect of salt upon gelation. I am not clear as to why two different salts were employed, since these seem to be discussed as being equivalent. It would certainly be ideal to use the same salt(s) for both of these assays rather than convoluting the argument by swapping one anion for another as this just leads to questions about robustness of the finding.

Reply: NaF was used for the CD spectroscopy experiments since chloride absorbs in the far UV and is therefore commonly avoided in such experiments. However, the reviewer is right in that the use of different anions can raise concerns. To ensure robustness of our findings we performed both CD spectroscopy and gelation experiments in the presence of both salts with very similar results. This data is now included in Fig 5 and Suppl Figs 2 and 8 and in the text on p 15.

5) The argument on p. 14 that elevated (NT monomer) stability is preventing formation of the hydrogel is quite elegant! To go beyond this and provide even more compelling demonstration of this behaviour, is the NT-L6 mutant also capable of gelation at elevated temperature? Given that NT* shows this behaviour, this is not by any means an essential experiment, but if these data were available that would certainly further bolster the argument.

Reply: We thank the reviewer for this suggestion and have now added data that shows that the NT-L6 mutant indeed forms a gel at 60 degrees (Fig 5c).

I also have several more minor suggestions for modifications/edits - noting that there are several other minor typos/wording issues which I have not explicitly mentioned that can certainly be corrected in revision. These are detailed in order of appearance:

1) Para. spanning pp. 2-3 – the discussion about spider silk behaviour (in solution, in the gland and in the fibre) is quite heavily based on major ampullate silk, which entirely makes sense based upon our level of understanding of this silk vs. other spider silks as well as the fact that the construct being used by the authors is MaSp1 based. But, this introductory section is worded in such a way that this could be interpreted by a reader as general

behaviour - I would suggest that it is best to ensure that the features and behaviour that are specific to major ampullate silk be noted as such.

Reply: We apologize for being unclear and have now improved the clarity of the introduction.

2) p. 3, para. 2 – 14 g/L is indeed an impressive yield, but “impressive” probably should be reworded.

Reply: “Impressive” has now been removed.

3) Para. spanning pp. 6-7 – the introductory sentence to this paragraph “Rheological measurements...” is not particularly informative, and does not provide a lot of context/lead-in for the next sentence which then introduces data. The types of moduli being measured and discussed should also be explicitly defined when these are first being reported, rather than in passing later in the paragraph. There is also some mixing of terms (storage/elastic vs. loss/viscous) in the figure captions, figures themselves, and main manuscript text (as well as later in Fig. 8/related discussion) which may overly convolute these studies. Finally, more detail about the rate of the temperature ramp may also be beneficial to include and discuss, especially with regard to its relation to the time-dependence observed for the gelation process.

Reply: We thank the reviewer for highlighting an area needing improvement for clarity and we apologize for the lack of consistency in our terminology. We have now amended this paragraph accordingly.

4) p. 8, para. 1 – would it be more appropriate to present these results on dope extracted from major ampullate gland before the rheology data? (It seems out of place in its current context.)

Reply: This is a good suggestion. The text has been modified accordingly.

5) p. 8, para. 2 (and Methods) – the FTIR data were acquired through ATR, but no mention of spectral correction for the effects of ATR is noted. Were any corrections applied for ATR – if so, this should be noted; if not, what is the rationale for not applying this correction to the data?

Reply: Indeed we show the original ATR spectra without correcting for the wavenumber dependence of the penetration depth. The reason for this is the small effect of the correction in the limited wavenumber interval that is of interest here: the penetration depth is only 9% larger at 1580 cm^{-1} than at 1720 cm^{-1} . We have added a figure (Supplementary Figure 4) which demonstrates that the corrected spectra deviate only very little from the original spectra. Because we do not make quantitative statements, we find a correction unnecessary. Even for quantitative secondary structure analysis, an ATR-FTIR spectra are usually used without correction. An example is one of the most prominent experts in that field, Erik Goormaghtigh, who does not correct his ATR spectra (e.g. Anal. Chem. 2021, 93, 3733–3741, BBA 2009, 1794, 1332–1343, Biophys. J. 2006, 90, 2946-2957). The BBA 2009 article compared CD, FTIR transmission and ATR-FTIR spectra of 45 proteins. The ATR-FTIR spectra were not corrected.

From these data, spectra of the "pure" secondary structures were extracted by linear regression and are shown in the figure below. The bottom spectra are those for "pure" α -helices and "pure" β -sheets. It can be seen that those obtained by ATR-FTIR spectroscopy (middle) are very similar to those obtained in transmission experiments. This similarity holds in spite of the different optical setups and in spite of the different hydration states (semi-dry film versus aqueous solution).

Fig. 6. Concatenated CD, ATR FTIR and transmission FTIR spectra representing the contribution of the 4 main secondary structures obtained by linear regression from the series of 45 concatenated spectra and the secondary structure content (see text).

6) p. 10, para. 1 (and Supp. Fig. 4) – the fluorescence micrographs are not entirely convincing as they appear to be entirely uniform over the entire micrograph – i.e., wouldn't at least some minimal fluctuations in intensity and/or dependence upon density of the material being imaged be expected? These may indeed be robust, but with such a complete lack of any discernable micrograph image features it leads one to wonder if there were technical issues with the instrument during acquisition and/or processing of the data.

Reply: We understand the reviewers concern that it is difficult to judge the diffuse green appearance of the gels in fluorescence microscopy images. However, this appearance is consistent with a homogeneous gel composed of ThT-positive nano-sized fibrils, and it was not possible to obtain higher resolution or better focus. Suppl. fig. 5b shows that the increase in ThT fluorescence occurs only after the gelation time point, which confirms that the green staining is not background fluorescence. To further support this point, we have now included images of gels that were formed in the presence of ThT, then broken to pieces and imaged in a TrisHCl buffer containing ThT (Suppl Fig 5c). There, we clearly see a contrast between ThT-positive gel pieces and the black background.

7) p. 10, para. 2 – I'm not convinced that the inset shown in Fig. 4c (a bulk integral over NH resonances) can be interpreted, as it has been, to show a uniform reduction of signal intensities. Instead, I would expect to see the attenuation of individual resonances analyzed and shown to be similar (within error) to demonstrate a "uniform" reduction.

Reply: We agree with the reviewer that the bulk integral might be misleading because it is mainly governed by the strongest signals. However, our analysis shows that in this case the

respective data for individual resonances (we added in the inset of Fig. 4c data for the same residues as in Fig. 4a, i.e., F24 and G136) exhibit a very similar trend as the bulk.

8) p. 11, para. 1, line 4 – the conclusion is made that “NT is most important for hydrogel formation”. Based upon the NMR analyses, this seems like a leap – i.e., certainly there’s demonstration that this region becomes structured etc., but stating that it’s “most important” seems to go beyond the current set of data and might be more appropriate at a later point in the manuscript?

Reply: We have removed this claim and replaced it with: “The results from NMR spectroscopy thus indicate that NT is important for hydrogel formation and convert into β -sheet conformation also when in fusion to 2Rep and CT”

9) p. 12, Fig. 4e – with the statement in the text (p. 11) that there’s predicted amyloid character in the five helices of the NT, it would be helpful for the positions of these helices with respect to the linear sequence to also be annotated/highlighted in some way.

Reply: Thank you for this suggestion. The position of the helices are now indicated in Fig 4e.

10) p. 14 and elsewhere – the NT^{L6} nomenclature is a little confusing, in that the same superscript style is also used to describe the single A82R mutant. I understand that the L6 mutant contains six Met -> Leu substitutions, but for ease of understanding I would suggest that a different way of referring to this mutant be used in order to ensure that it’s not thought to be referring (albeit incompletely!) to a single mutation at residue 6.

Reply: We agree and have now changed the naming of this mutant to His-NT-L6.

11) p. 15, panels b, d and f – are there representative CD spectra for each construct to back up these inferences about folding that are based on temperature scan measurement at only 222 nm? I would be concerned that without this context the ellipticity at 222 nm alone is not diagnostic of the “folded state” of the protein. These data certainly do not need to be in the main manuscript, but could be provided in the Supp. Info.

Reply: CD spectra for each construct at 25°C and 95°C have been added as Suppl. Fig 8

12) p. 16, para. 1 – ensure that species names are italicized and note that “Supplementary” Table 2 is misspelled in the last sentence. I would also suggest that the final sentence would benefit from a little more detailing about what, exactly, the correlation being proposed is.

Reply: Thanks for pointing this out. We have corrected these mistakes.

13) p. 16, para. 2 – the second sentence would benefit from being reworded (i.e., it currently has a mix of gene- vs. protein-oriented descriptions).

Reply: This is now corrected.

14) Para. spanning pp. 16-17 and Fig 6b – it's not fully clear that the PNP fusion is being "diluted", nor why. The figure caption also implies that this is a sample composed entirely of PNP fusion, which is inconsistent with the text. In terms of the dilution, a reader might be left wondering if this is needed due to an issue with the enzyme remaining functional as a fusion protein if it's undiluted. I recognize that this is mentioned in more depth in the methods section (with the argument, as I read it, that activity was "too high" to assay), but it would be valuable to detail this up front as part of the results section. I am also curious how the values of U/mg were determined – from the manner in which the data are presented, this could be interpreted as being known "a priori" but presumably this was actually determined through the fluorogenic assay?

Reply: We apologize for being unclear. We hope that the added text in the figure legend of Fig 6 and corresponding results section makes the experimental set-up and reason for dilution of the fusion protein clear.

15) p. 20, para. 1 – the arguments being made would also benefit from discussion of the anticipated temperature in the gland itself. (i.e., if a temp. of ~37 °C is needed for effective gelation, how does this compare to the temperature in the gland? Presumably that's mostly environmentally determined for a given spider?)

Reply: This is an interesting point that deserves to be investigated in future studies. As the reviewer points out, the temperature in the gland is likely the same as the one in the spider's environment, but there are many additional factors that will affect the gelation of the spidroins (as briefly discussed on p 21). In addition to the points raised in the manuscript, spider species, protein concentration and composition of the spinning dope will likely influence the gelation of the spidroins. Furthermore, since we study recombinant miniature spidroins and isolated domains and not full length spidroins, it would be highly speculative to add such a discussion. We therefore prefer to refrain from further speculation at this point.

16) pp. 23-26 – the protein sequences are indeed essential and the format in which they are provided is ideal, with respect to being able to be copied and pasted. However, these would likely be better provided either in a table in the main text or in the Supporting Information, rather than as a standalone segment of the Methods section.

Reply: The amino acid sequences have now been moved to a new supplementary table 4.

In support of the desire by Nature Communications to increase transparency, I am happy to sign this review:

Jan K. Rainey, Ph.D.

Department of Biochemistry & Molecular Biology

Dalhousie University

Halifax, Nova Scotia, Canada

Reviewer #2 (Remarks to the Author):

The authors report the N-terminal domain (NT) and its fusion proteins form self-supporting and transparent hydrogels at 37 °C, and thus provide a protein immobilization platform. In addition, they also explore the gelation mechanism and find the gelation is caused by NT α -helix to β -sheet conversion and formation of amyloid-like fibrils. The study may be of general interest but enthusiasm is limited by a lack of novelty and concerns about the gelation mechanism.

Reply: We thank the reviewer for their efforts and apologize for the apparent lack of clarity with respect to conveying the novelty and impact of the work. We hope the changes made during revision will make novelty of our data more clear.

Major concerns

1. Novelty. It is not surprising to see the proteins or peptides forming self-supporting hydrogels at 37 °C although this is the first time to be seen for NT. For example, elastin-like proteins, silk-elastin-like proteins and many other designed polypeptides could form temperature-responsive hydrogels and have been demonstrated to be used in drug delivery and tissue engineering. The gelation mechanism for these proteins and peptides is also widely explored.

Reply: The novelty in our findings lie in the NT, which is a folded globular protein, known for its solubility and ability to refold, turns into β -sheet fibril gels at 37 °C. This is different from the mentioned silk derived proteins since they typically adopt an unstructured conformation. Furthermore, since NT can be employed as an expression tag to acquire high protein yields, the system can be used for both producing soluble fusion proteins at high yields and turn them into hydrogels with intact functionality of the fusion partner. This combination of properties has not been reported before.

This is pointed out in the first and last paragraph of the discussion and the difference between our hydrogels and similar gels previously reported in the literature are shown in Fig 7.

2. As we know, the NTs are evolutionarily conserved within spidroins of different spider species, why do only some of them form hydrogels? Does the gelation of NT contribute to spider silk formation? This might be more interesting than gelation itself.

Reply: This is an interesting point but as mentioned above, the behavior of NT in isolation and at the concentrations used for in vitro gelation may not be related to NT's role in silk spinning in the spider. Determination of the importance of our observations for fiber spinning will require extensive work, see reply to Reviewer #1, question 15.

3. As shown in Figure 1b, the gelation of NT is time-dependent. The authors should study the evolution of transparency, secondary structures, internal morphologies of hydrogels with time, to explore the gelation mechanism.

Reply: The manuscript already contain this information, and the results are shown in the current Suppl Fig 5, Fig 3f, and in Fig 5d,f.

4. As shown in Figure 2b and 2c, the gelation temperature (cross-over point when $G' > G''$) for His-NT2RepCT (300mg/ml) is around 40°C, but for NT (300mg/ml) is below 25°C. This result seems inconsistent with the findings by vial inversion in figure 1b and c. The authors should identify the gelation temperature by oscillatory rheology first and then explore the evolving of gels with time at a specific temperature.

Reply: We apologize for the confusion here. As a result of discussion between the authors and a reanalysis of the data, as a precaution we have removed the loss modulus data from this figure. This is to improve clarity of the results, aid comparison to figure 1 and avoid interpreting a crossover as a defined gel point due to the low modulus of these samples and it occurring near an area in which the minimum torque values of the rheometer were approached (based on manufacturers specifications). However, for completeness we do include and signpost the reader to all the original data and more in supplementary figure 3.

Regarding the identification of gelation temperature, this is a good point and something that sparked a lot of discussion between the authors at the time of experimentation. Here, we opted for a comparable rate to our previous studies (1 °C/min, Holland, C. *et al Nat. Mat.* 2006 5, 870-874. Laity, P *et al Int. J. Mol. Sci.* 2016, 17, 1812) and will be focusing on exploring the extended gelation kinetics of these materials in future work.

Minor points:

5. Page 10, the authors point that “but addition of 5, 10 or 20 % (w/w) NT hydrogels to NT solutions did not give any pronounced seeding effect (Fig. 3g).” The seeds are usually nanofibrils in solutions, how could the authors use the hydrogels as seeds? The fibrils in the hydrogels are relatively fixed and not easy to be reached as seeds.

Reply: We thank the reviewer for raising this important point. We have added this possible explanation to the results section.

“Possibly, this could be due to that the fibrils in the hydrogels are relatively fixed and may be inaccessible for acting as seeds.”

6. In figure 2, the protein concentration used for turbidity measurement is 100mg/ml, but for oscillatory measurement is 300mg/ml, why not be consistent?

Reply: Thank you for highlighting this observation. Oscillatory measurements were done at 100 as well as 300 and 500 mg/ml (Supplementary Figure 3). The point of including the turbidity measurements in this work was to highlight that the increase in turbidity (when observed) is not correlated to the increase in ThT fluorescence. To improve the readability and clarity of the manuscript we now moved figure 2A to the supplementary section (now Suppl Fig 5d) and refer to it when discussing the results from the ThT assay on p. 11.

7. In figure 1c and figure 2d, replicates are not included.

Reply: Thank you for bringing this to our attention. Replicates are now indicated in the figures.

8. Page 16, “The His-NT-GFP (300 mg/ml) and His-NT-PNP (100 mg/ml) fusion proteins formed gels after 2 and 6.5 hours at 37 °C”. From the application’s point of view, the gelation concentrations are relatively high and the gelation time is long. Please discuss how to improve this for the application.

Reply: We are not sure which application that the reviewer refers to. As shown in Fig 7, our gels have unique characteristics and fast gelation when compared to previously described recombinant spider silk gels and commonly used gels made from other proteins and polysaccharides.

9. The storage modulus and loss modulus in figure 2b and c should be corrected to the elastic modulus and viscous modulus, to be consistent with the figure legend.

Reply: This has now been corrected.

10. Ref 25 and 39 are same papers, 77 and 84 are same.

Reply: Thank you for spotting this. It has now been corrected.

Reviewer #3 (Remarks to the Author):

This manuscript claims that recombinant N-terminal domain (NT) from spidroins can rapidly form self-supporting and transparent hydrogels at 37 °C, and also provide a protein immobilization platform that can introduce gelation when fused with other protein. Overall, most of the results support the authors’ conclusions and claims. However, I recommend that the manuscript should fully address the following questions:

Reply: We thank the reviewer for their time on our work and hope that the below response is to their satisfaction.

1. The gelation of recombinant spider silk proteins is very common in lots of other researches, so what is the significance in this article?

Reply: We thank the reviewer for their efforts and apologize for the apparent lack of clarity with respect to conveying the novelty and impact of the work. We hope the changes made during revision will make novelty of our data clearer.

2. The authors mentioned that “All hydrogels, except 2RepCT gels, were optically transparent to the naked eye (Fig. 1b). This was verified for all gels by measuring light transmission, with

exception of NT2Rep and CT gels which showed an increase in absorbance at 360 nm over time (Fig. 2a).”

As presented in Figure 2b, I think not all the hydrogels were optically transparent, for example, HIS-NT2RepCT, NTCT. Also, the authors claimed optically transparent, but why the authors only measured the light absorbance at 360 nm? To my knowledge, 360 nm is in the range of UV light, rather than in the visible light ranged from 380-780 nm.

Reply: We thank the reviewer for pointing this out and apologize for not being precise in our descriptions. “Absorbance” at 360 is traditionally used to measure turbidity, ie it is not absorbance but light scattering due to particles that is measured. The paragraph on p 5 now reads:

“The hydrogels formed from the different recombinant spidroins had slightly different colors and showed different degree of transparency as judged by the naked eye (**Error! Reference source not found.b**). The NT gels were exceptionally clear while others became opaque. His-NT2RepCT and NT gels that were cast in cylindrical tubes could be removed intact from the molds (**Error! Reference source not found.d**).”

Furthermore, we moved Fig 2A to the supplement (now Suppl Fig 5d) and discuss it in relation to the results from the ThT experiments on page 11.

3. The authors mentioned that “The band at 1617 cm^{-1} was more pronounced for NT than for His-NT2RepCT, suggesting a higher overall beta-sheet content in NT hydrogels than in NT2RepCT hydrogels.”

I recommend the authors to revise this statement. First, deconvolution of the peaks is necessary if the authors want to discuss the content of beta-sheet. Then, the ATR model is not favorable for the deconvolution of the specific peaks because lights in this model do not go through the whole sample, that is not following the beer-lambert law. The surface transparency of the samples will significantly affect their absorbance peaks.

Reply: The reviewer is correct in pointing out that the IR light does not penetrate through the entire specimen, in the attenuated total reflection (ATR) configuration, with the evanescent wave decaying exponentially over a few micrometres from the surface of the ATR element, as described by the Harrick equation [1,2]. Nevertheless, when correction for penetration depth was applied (see the new Supplementary Figure 4), its effects were hardly visible and consequently, the IR interaction with the specimen is expected to be relatively uniform across the spectral window of interest (1580-1720 cm^{-1} , for the amide I band).

In view of that, it is anticipated that quantitative (or at least semi-quantitative) analysis of the components within the amide I band relating to different structural features in the protein is valid, as demonstrated by previous work (e.g. see: Percot et al. [3]). Thus ATR-FTIR spectroscopy is commonly used for secondary structure analysis (see e.g. the comprehensive work by Erik Goormaghtigh).

In order to strengthen our argument, we have deconvoluted our spectra by calculating the second derivative of absorbance, which reveals component bands better than the absorbance spectra (Supplementary Figure 4b). Gel formation clearly leads to the appearance of high and low wavenumber β -sheet bands which are largely absent before gelation. Additionally, we

have calculated difference spectra which reveal the spectral changes upon gel formation (Supplementary Figure 4a). The difference spectra demonstrate an increase in absorption associated with β -sheets and a decrease of absorption related to α -helices. Thus, both analyses support our view of a structural change from α -helix to β -sheets upon gelation and we hope that - in the light of these additional tests - the reviewer can now agree with our interpretation.

1. Harrick, NJ. Surface chemistry from spectral analysis of totally internally reflected radiation, *J. Phys. Chem.* 1960, *64*, 1110–1114, <https://doi.org/10.1021/j100838a005>
2. Buffeteau, T. Desbat, B. Eyquem, D. Attenuated total reflection Fourier transform infrared microspectroscopy: Theory and application to polymer samples, *Vibr. Spectr.* 1996, *11*, 29-36, [https://doi.org/10.1016/0924-2031\(95\)00054-2](https://doi.org/10.1016/0924-2031(95)00054-2)
3. Percot, A. Colomban, P. Paris, C. Dinh, HM. Wojcieszak, M. Mauchamp, B. Water dependent structural changes of silk from *Bombyx mori* gland to fibre as evidenced by Raman and IR spectroscopies, *Vibr. Spectr.* 2014, *73*, 79-89, <http://dx.doi.org/10.1016/j.vibspec.2014.05.004>

4. The authors mentioned that “Analysis of the gel by transmission electron microscopy (TEM) revealed that the hydrogels are composed of amyloid-like fibrils (Fig. 3c-d).” Do the authors check the orientation of the beta-sheet in this nanofibrils? Cross-beta or not is the difference between the amyloid-like fibrils and the silk fibroin-like fibrils.

Reply: We refer to the fibrils as amyloid-like since they show morphological features and ThT binding that are two characteristic features of amyloid fibrils. How these fibrils relate to the fibrils observed in the silk fiber remains to be elucidated.

5. More details should be provided for the hydrogels of the fusion proteins. For example, how to express the proteins and to form hydrogels? Do the authors run the PAGE to check the expression? Do the authors consider the free proteins in the hydrogels and solution?

Reply: This information is given in detail in the materials and methods section, see “protein expression and purification” and “gelation of proteins”

REVIEWERS' COMMENTS

Reviewer #1 (Remarks to the Author):

The authors have extremely comprehensively and satisfactorily addressed all of my questions, concerns and comments during the revision process. I now feel the manuscript to be fully suitable for publication.

Jan K. Rainey, Ph.D.
Department of Biochemistry & Molecular Biology
Dalhousie University
Halifax, Nova Scotia, Canada

Reviewer #2 (Remarks to the Author):

The authors have addressed most of my concerns. There are only some minor comments that should be addressed.

To clarify the novelty, the authors supplemented Fig 7. Please make sure that the determination methods for gelation are similar between the NT gels and other reported ones, otherwise they are not comparable. In addition, please add the gels formed from elastin-like proteins, silk-elatin-like proteins etc., which are more appropriate than the alginate and agar gels. You may read some elastin references such as *Expert Opinion on Drug Delivery*, 14:1, 37-48; *Biomacromolecules* 2015, 16, 3762–3773; *ACS Macro Lett.* 2021, 10, 395–400.

REPLY TO REVIEWERS

Reviewer #1 (Remarks to the Author):

The authors have extremely comprehensively and satisfactorily addressed all of my questions, concerns and comments during the revision process. I now feel the manuscript to be fully suitable for publication.

Jan K. Rainey, Ph.D.
Department of Biochemistry & Molecular Biology
Dalhousie University
Halifax, Nova Scotia, Canada

Reply: We thank Dr Rainey for the time and effort spent on improving our manuscript.

Reviewer #2 (Remarks to the Author):

The authors have addressed most of my concerns. There are only some minor comments that should be addressed.

To clarify the novelty, the authors supplemented Fig 7. Please make sure that the determination methods for gelation are similar between the NT gels and other reported ones, otherwise they are not comparable. In addition, please add the gels formed from elastin-like proteins, silk-elatin-like proteins etc., which are more appropriate than the alginate and agar gels. You may read some elastin references such as Expert Opinion on Drug Delivery, 14:1, 37-48; Biomacromolecules 2015, 16, 3762–3773; ACS Macro Lett. 2021, 10, 395–400.

Reply: We thank the reviewer for suggesting further improvements of the manuscript. We have now indicated which method was used for determining the time of gelation in the Supplementary Table 1, and we have added the suggested information in Suppl Table 3 and in Fig 7.